# FlowCast: Advancing Precipitation Nowcasting with Conditional Flow Matching

**Bernardo Perrone Ribeiro**
University of Ljubljana
brbmp0@gmail.com

**Jana Faganeli Pucer**
University of Ljubljana
jana.faganelipucer@fri.uni-lj.si

## Abstract

Radar-based precipitation nowcasting, the task of forecasting short-term precipitation fields from previous radar images, is a critical problem for flood risk management and decision-making. While deep learning has substantially advanced this field, two challenges remain fundamental: the uncertainty of atmospheric dynamics and the efficient modeling of high-dimensional data. Diffusion models have shown strong promise by producing sharp, reliable forecasts, but their iterative sampling process is computationally prohibitive for time-critical applications. We introduce FlowCast, the first end-to-end probabilistic model leveraging Conditional Flow Matching (CFM) as a direct noise-to-data generative framework for precipitation nowcasting. Unlike hybrid approaches, FlowCast learns a direct noise-to-data mapping in a compressed latent space, enabling rapid, high-fidelity sample generation. Our experiments demonstrate that FlowCast establishes a new state-of-the-art in probabilistic performance while also exceeding deterministic baselines in predictive accuracy. A direct comparison further reveals the CFM objective is both more accurate and significantly more efficient than a diffusion objective on the same architecture, maintaining high performance with significantly fewer sampling steps. This work positions CFM as a powerful and practical alternative for high-dimensional spatiotemporal forecasting. [1]

## 1 Introduction

Accurate and timely short-term precipitation forecasts, or nowcasting, are of paramount importance due to their significant socio-economic impacts, such as issuing flood warnings and managing water resources. Precipitation nowcasting, as defined in this work, involves predicting a sequence of future radar images from historical observations for the immediate future up to a few hours (An et al., 2025). Traditional methods, like Eulerian and Lagrangian persistence (Germann & Zawadzki, 2002), rely on advecting the current precipitation field. However, their simplified physical assumptions limit their ability to capture the complex, non-linear dynamics of atmospheric processes, especially for rapidly evolving weather systems (Prudden et al., 2020).

Deep learning has introduced a paradigm shift in precipitation nowcasting. Deterministic models based on recurrent and transformer architectures learn complex spatiotemporal patterns directly from large volumes of radar data (Prudden et al., 2020; An et al., 2025). While these models outperform traditional methods, optimizing for metrics like Mean Squared Error (MSE) compels them to produce a single, best-guess forecast. This often results in overly smooth predictions at longer lead times, failing to capture the inherent uncertainty in precipitation evolution and underrepresenting high-impact weather events.

To address this, probabilistic generative models have become central to modern nowcasting, aiming to predict a distribution over many plausible futures. Diffusion models (Ho et al., 2020), in particular, have emerged as the state-of-the-art, producing sharp and reliable ensemble forecasts (Gao et al., 2023; Leinonen et al., 2023; Gong et al., 2024). However, their reliance on an iterative denoising process, often requiring hundreds of function evaluations for a single forecast, makes them computationally expensive. This high Number of Function Evaluations (NFE) poses a significant barrier in time-critical scenarios where rapid ensemble generation is crucial.

---

[1]Our implementation is available at https://github.com/b-rbmp/FlowCast.

This work introduces FlowCast, a novel probabilistic nowcasting model built on Conditional Flow Matching (CFM) (Lipman et al., 2023; Tong et al., 2024), a powerful and efficient alternative designed for rapid sampling. While recent work has applied rectified flows for the deterministic refinement of blurry forecasts (Feng et al., 2025), FlowCast is, to our knowledge, the first to successfully apply CFM as a full, noise-to-data generative model for this task. We demonstrate that FlowCast alleviates the tension between accuracy and efficiency, establishing a new state-of-the-art by exceeding the performance of leading diffusion models while offering a superior performance-cost trade-off.

We argue that CFM offers not only a computational advantage but also a superior inductive bias for this domain, specifically regarding the simplified transport of probability mass. Radar reflectivity distributions are highly multi-modal yet exhibit strong local temporal consistency. Standard diffusion models map Gaussian noise to this complex manifold via stochastic denoising or curved probability flow ODEs, often necessitating many sampling steps to resolve fine-grained structures without blurring modes. In contrast, CFM imposes a straight-line ODE prior on the generative process. This enforces the simplest possible mapping between the noise and data distributions. In the context of spatiotemporal forecasting, where temporal coherence is essential, this linear interpolation provides a much stronger and more stable prior than the winding paths of diffusion. We demonstrate that this geometric simplification allows FlowCast to maintain high fidelity with significantly fewer function evaluations.

Our contributions are summarized as follows:

- We introduce FlowCast, a novel full-probabilistic application of Conditional Flow Matching to precipitation nowcasting.
- We establish a new state-of-the-art in both probabilistic performance and predictive accuracy on two diverse radar datasets, the benchmark SEVIR dataset (Veillette et al., 2020) and the local ARSO dataset.
- We provide a direct ablation study showing that the CFM objective is both more accurate and more computationally efficient than a diffusion objective on the same architecture, maintaining high performance with substantially fewer sampling steps.

## 2 RELATED WORK

### 2.1 DETERMINISTIC NOWCASTING

Deep learning for precipitation nowcasting has evolved from RNN-based architectures to Transformer-based models. Early work includes ConvLSTM (Shi et al., 2015), extending LSTMs with convolutions for spatiotemporal data, and the PredRNN family (Wang et al., 2017; 2023), which introduced a spatiotemporal memory flow for improved long-range dependency modeling. More recently, Transformer architectures like Earthformer (Gao et al., 2022) and Earthfarseer (Wu et al., 2024) have set new benchmarks by using attention to model complex global dynamics. A common limitation of deterministic models is that they produce overly smooth forecasts when trained with pixel-wise losses (e.g., MSE), as they average over possible futures.

### 2.2 PROBABILISTIC NOWCASTING

To address uncertainty quantification, probabilistic models have become central to nowcasting, aiming to sample from the full distribution of future states.

**GANs and Diffusion.** GANs (Ravuri et al., 2021) were an early approach for producing sharp forecasts but suffer from training instability. Diffusion models (Ho et al., 2020) have recently emerged as the state-of-the-art, offering stable training and high-quality samples. PreDiff (Gao et al., 2023) and LDCast (Leinonen et al., 2023) are prominent latent diffusion models for ensemble forecasting. A notable hybrid is CasCast (Gong et al., 2024), which uses a deterministic model for large-scale patterns and a conditional diffusion model to refine stochastic details.

**Flow-Based Generative Models.** Generative modeling with flows offers an attractive alternative to diffusion. Traditional Continuous Normalizing Flows (CNFs) (Chen et al., 2018) model data

via ODEs but require expensive numerical integration during training to compute likelihoods, making them computationally prohibitive for high-dimensional spatiotemporal data. Conditional Flow Matching (CFM) (Lipman et al., 2023) and Rectified Flows (Liu et al., 2023) overcome this by regressing a vector field against a conditional probability path, enabling simulation-free training. However, while standard Rectified Flows typically utilize a singular conditional path (effectively $\sigma \to 0$), our application of Independent CFM (I-CFM) incorporates a Gaussian probability path with $\sigma > 0$. This "thickens" the training trajectory, providing crucial regularization that stabilizes the learning of the vector field for high-dimensional data compared to the singular paths of rectified flows.

Crucially, while diffusion models rely on stochastic denoising paths that are often curved and require many sampling steps, CFM allows for learning straight-line ODE trajectories between noise and data (Tong et al., 2024). This geometric property enforces a direct mapping that preserves temporal coherence and allows for rapid sampling. While Feng et al. (2025) recently used a rectified flow module to strictly refine deterministic forecasts, FlowCast applies CFM as a standalone probabilistic generative model. This allows it to learn the full noise-to-data distribution and capture multimodal uncertainty without relying on a deterministic base forecast.

## 3 METHOD

Our approach to probabilistic nowcasting is based on Conditional Flow Matching (CFM) within a compressed latent space. This section details our methodology, covering the problem formulation, our latent CFM framework, the model architecture, and the training and sampling procedures.

### 3.1 TASK FORMULATION

Precipitation nowcasting is framed as a video prediction task. Given a sequence of $T_{\text{in}}$ past radar observations, $\mathbf{X}_{\text{past}} = \{x_1, x_2, \ldots, x_{T_{\text{in}}}\}$, where each $x_t \in \mathbb{R}^{H \times W \times C}$ is a radar map, the objective is to generate a probabilistic forecast for the next $T_{\text{out}}$ frames, $\mathbf{X}_{\text{future}} = \{x_{T_{\text{in}}+1}, \ldots, x_{T_{\text{in}}+T_{\text{out}}}\}$.

### 3.2 LATENT CONDITIONAL FLOW MATCHING

To reduce the high computational cost of generative modeling, we adopt a two-stage approach inspired by latent diffusion models (Rombach et al., 2022). A Variational Autoencoder (VAE) (Kingma & Welling, 2014) compresses high-dimensional radar frames into low-dimensional latents, which are used to train a generative model in the latent space.

Our generative model is built on the Conditional Flow Matching (CFM) framework (Lipman et al., 2023), which trains a continuous normalizing flow by learning a vector field $v_\theta$ that maps samples from a prior distribution (e.g., Gaussian) to the target data distribution. We use Independent CFM (I-CFM) (Tong et al., 2024), which defines a probability path $p_t(x_t|x_0, x_1)$ as a Gaussian distribution with mean $(1-t)x_0 + tx_1$ and a small constant standard deviation $\sigma$. This path interpolates between a noise sample $x_0 \sim \mathcal{N}(0, I)$ and a data sample $x_1$. The corresponding target vector field is their difference, $u_t = x_1 - x_0$. This formulation enables direct, simulation-free training of the model $v_\theta$ by regressing it against this target field.

#### 3.2.1 FRAME-WISE AUTOENCODER

To learn a compact latent space, we train a VAE on individual radar frames. The architecture, inspired by Esser et al. (2021), uses a hierarchical encoder $\mathcal{E}$ and decoder $\mathcal{D}$ with residual and self-attention blocks for high-fidelity reconstructions. The VAE is trained with a combination of a L1 reconstruction loss, a KL-divergence regularizer, and a PatchGAN adversarial loss (Isola et al., 2017) to enhance perceptual quality. After training, the VAE's weights are frozen and it is used to encode inputs and decode latent predictions.

#### 3.2.2 FLOWCAST ARCHITECTURE

We propose **FlowCast**, which consists of the adaptation of Earthformer-UNet (Gao et al., 2023) for the CFM objective. FlowCast employs a U-Net-like encoder-decoder structure where the core

building blocks are Cuboid Attention layers from Earthformer (Gao et al., 2022). This mechanism efficiently processes spatiotemporal data by applying self-attention locally within 3D "cuboids" of the data, capturing local dynamics, while global information is shared across the hierarchical U-Net structure. The model is conditioned on the flow time $t$, which is converted into an embedding and injected at each level of the network, enabling the model to accurately approximate the time-dependent vector field $v_\theta$. The architecture is illustrated in Figure 1.

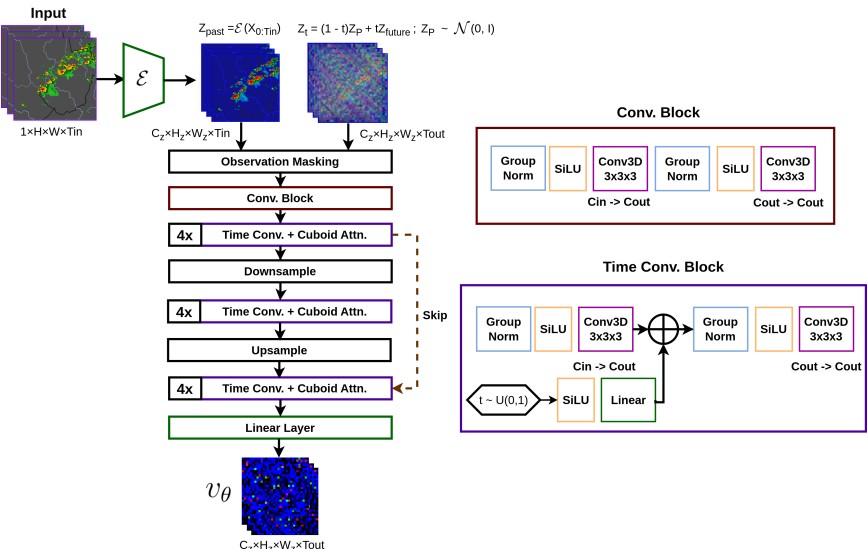

Figure 1: The FlowCast architecture. A U-Net with Cuboid Attention blocks processes latent spatiotemporal data. Conditioning on the flow time $t$ enables the model to learn the time-dependent vector field for generating forecasts.

**Training.** The FlowCast model learns the vector field $v_\theta(Z_t, t, Z_{\text{past}})$. The complete training procedure is detailed in Algorithm 1.

---

**Algorithm 1** FlowCast Training Process

---

**Require:** Dataset $\mathcal{D}$, Pre-trained VAE Encoder $\mathcal{E}$, FlowCast Model $v_\theta$, standard deviation $\sigma$.
  1: **Initialize** model parameters $\theta$
  2: **while** not converged **do**
  3:     Sample a batch of radar sequences $(X_{\text{past}}, X_{\text{future}}) \sim \mathcal{D}$
  4:     Encode sequences into latent space: $Z_{\text{past}} \leftarrow \mathcal{E}(X_{\text{past}})$ and $Z_{\text{future}} \leftarrow \mathcal{E}(X_{\text{future}})$
  5:     Sample prior noise $Z_P \sim \mathcal{N}(0, I)$, time $t \sim \mathcal{U}(0, 1)$ and path Gaussian noise $\epsilon \sim \mathcal{N}(0, I)$
  6:     Compute interpolated latent state: $Z_t \leftarrow (1 - t)Z_P + tZ_{\text{future}} + \sigma\epsilon$
  7:     Compute target vector field: $u_t \leftarrow Z_{\text{future}} - Z_P$
  8:     Predict vector field: $\hat{v} \leftarrow v_\theta(Z_t, t, Z_{\text{past}})$
  9:     Compute Loss: $\mathcal{L} \leftarrow ||\hat{v} - u_t||^2$
 10:     Gradient step: $\theta \leftarrow \theta - \eta\nabla_\theta\mathcal{L}$
 11: **end while**

---

**Sampling.** To generate an ensemble of forecasts, we solve the learned ODE starting from noise, using the Euler method (Hairer et al., 1993). This process is described in Algorithm 2.

## 4 EXPERIMENTS

### 4.1 EXPERIMENTAL SETTING

More details about the experimental setting are provided in Appendix A.1.

---

**Algorithm 2** FlowCast Ensemble Sampling

---

**Require:** Past radar sequence $X_{\text{past}}$, Trained FlowCast Model $v_\theta$, VAE Encoder $\mathcal{E}$ / Decoder $\mathcal{D}$.
**Require:** Ensemble size $N$, Number of ODE steps $S$.
**Ensure:** Ensemble forecasts $\{\hat{X}_{\text{future}}^{(k)}\}_{k=1}^N$.
 1: Encode past observations: $Z_{\text{past}} \leftarrow \mathcal{E}(X_{\text{past}})$
 2: **for** $k = 1$ **to** $N$ **do**
 3:     Sample initial state: $Z(0) \sim \mathcal{N}(0, I)$
 4:     Set step size: $\Delta t \leftarrow 1/S$
 5:     Set current state: $Z_{curr} \leftarrow Z(0)$
 6:     **for** $i = 0$ **to** $S - 1$ **do**
 7:         Current time: $t \leftarrow i \cdot \Delta t$
 8:         Predict vector field: $v \leftarrow v_\theta(Z_{curr}, t, Z_{\text{past}})$
 9:         Update state (Euler step): $Z_{curr} \leftarrow Z_{curr} + v \cdot \Delta t$
10:     **end for**
11:     Final latent forecast: $Z(1) \leftarrow Z_{curr}$
12:     Decode to pixel space: $\hat{X}_{\text{future}}^{(k)} \leftarrow \mathcal{D}(Z(1))$
13: **end for**
14: **return** Ensemble predictions $\{\hat{X}_{\text{future}}^{(k)}\}_{k=1}^N$

---

### 4.1.1 DATASETS

We evaluate on two 5-minute, 1 km-resolution radar datasets: SEVIR, a US benchmark, and ARSO, a Slovenian composite for a local deployment setting. For both, we predict 12 frames (1 hour) from 13 past frames (65 minutes), per the SEVIR Nowcasting Challenge protocol (Veillette et al., 2020).

Table 1: Summary of datasets used for evaluation.

| Dataset | $N_{\text{train}}$ | $N_{\text{val}}$ | $N_{\text{test}}$ | Resolution | Dimensionality | Interval | Lag/Lead |
|---------|-------|-------|-------|------------|----------------|----------|----------|
| SEVIR   | 36,351 | 9,450 | 12,420 | 1 km | 384×384 | 5 min | 13/12 |
| ARSO    | 38,229 | 12,743 | 12,744 | 1 km | 301×401 | 5 min | 13/12 |

**SEVIR.** SEVIR (Veillette et al., 2020) provides over 10,000 weather events in a 384×384 km US domain, each spanning 4 hours at 5-minute resolution. We use the 1-km Vertically Integrated Liquid (VIL) field. Following the standard chronological split, we extract 25-frame sequences (13 context, 12 target) with a stride of 12, yielding 36,351 training, 9,450 validation, and 12,420 test samples.

**ARSO.** The ARSO dataset contains 5-minute, 1-km radar reflectivity composites over a 301×401 km Slovenian grid, capturing complex Alpine and coastal dynamics. Using the same 25-frame sequence setup but with stride 1, a 60/20/20 chronological split yields 38,229 training, 12,743 validation, and 12,744 test samples.

### 4.1.2 EVALUATION

**Threshold-based categorical scores:** Following prior work (Veillette et al., 2020; Gao et al., 2023; Gong et al., 2024), we evaluate forecasts by converting radar fields to binary masks at given thresholds and computing the False Alarm Ratio (FAR), Critical Success Index (CSI), and Heidke Skill Score (HSS). For spatial validation, we compute the max-pooled CSI and Fractions Skill Score (FSS) over $16 \times 16$ km neighborhoods (CSI-M-P16 and FSS-M-P16). We report the mean of these scores across all thresholds ("-M") to evaluate general performance. Furthermore, to rigorously assess the detection of extreme weather events, we separately report the categorical metrics specifically at the highest intensity thresholds for each dataset.

For SEVIR, we follow the literature in using the thresholds $[16, 74, 133, 160, 181, 219]$. For ARSO, we use the thresholds $[15, 21, 30, 33, 36, 39]$ dBZ, derived through quantile mapping to ensure that each threshold corresponds to approximately the same exceedance probability in both datasets.

**Continuous Ranked Probability Score (CRPS):** We use the CRPS to evaluate probabilistic skill. A lower CRPS indicates a more accurate and sharp forecast. For deterministic forecasts ($N = 1$), CRPS reduces to the Mean Absolute Error.

**Ensemble forecasting:** Let $x_{t,i,j}$ represent the ground truth pixel value at location $(i, j)$ and lead time $t$. All probabilistic models are evaluated using an ensemble of $N = 8$ realizations. For categorical scores, we evaluate the ensemble mean prediction (Metric of Ensemble Mean), first computing the ensemble mean $\hat{x}_{t,i,j} = \frac{1}{N} \sum_{k=1}^{N} \hat{x}_{t,i,j}^{(k)}$ and then the metric on this mean forecast.

### 4.1.3 TRAINING DETAILS

**VAE.** We train a separate Variational Autoencoder (VAE) for each dataset to create a specialized latent space. We follow the architecture and training procedure from Rombach et al. (2022), with a Kullback-Leibler divergence loss weight of 1e-4, the AdamW optimizer with a learning rate of 1e-4, and a batch size of 12. The compressed latent space dimensions are shown in Table 2.

Table 2: VAE latent space dimensions

| Dataset | Original Dimensions $(T_{\text{in}}/T_{\text{out}} \times \mathbf{H} \times \mathbf{W} \times \mathbf{C}_{\text{in}})$ | Latent Dimensions $(T_{\text{in}}/T_{\text{out}} \times \mathbf{H}_z \times \mathbf{W}_z \times \mathbf{C}_z)$ |
|---|---|---|
| SEVIR | $13/12 \times 384 \times 384 \times 1$ | $13/12 \times 48 \times 48 \times 4$ |
| ARSO | $13/12 \times 301 \times 401 \times 1$ | $13/12 \times 38 \times 52 \times 4$ |

**FlowCast.** We train our CFM model for 200 epochs using the AdamW optimizer with a learning rate of 5e-4 and a cosine scheduler. We set the standard deviation of the I-CFM probability path to a small constant $\sigma = 0.01$ (Tong et al., 2024). We observed that the training process was notably stable; unlike diffusion objectives which often require complex loss weighting schedules, the I-CFM objective utilizes a simple regression loss that converged robustly without extensive hyperparameter tuning. Model checkpoints are maintained using an exponential moving average of weights (Ho et al., 2020), with a decay factor of 0.999, and we keep the model checkpoint with the highest CSI-M evaluated on a subset of the validation set. The model is trained with 4 NVIDIA H100 for 7 days, with a global batch size of 12. Further implementation details are provided in Appendix A.1.

### 4.1.4 INFERENCE DETAILS

Generating a forecast with FlowCast involves solving the learned ODE to transform a noise-initialized latent sequence into a prediction, conditioned on encoded past observations. Following the procedure outlined in 3.2.2, we use the Euler method (Hairer et al., 1993) with 10 steps as the ODE solver. To generate a probabilistic ensemble forecast, this process is repeated eight times with different initial noise samples $Z(0) \sim \mathcal{N}(0, \mathbf{I})$.

### 4.2 COMPARISON TO THE STATE OF THE ART

We evaluate FlowCast against four deterministic baselines: U-Net (Veillette et al., 2020), Earthformer (Gao et al., 2022), Earthfarseer (Wu et al., 2024), and SimVPv2 (Tan et al., 2025), as well as two probabilistic baselines: PreDiff (Gao et al., 2023) and CasCast (Gong et al., 2024). All models are trained following their publicly released code, with the training budget fixed at 200 epochs. For probabilistic models, we adopt our evaluation protocol by selecting the checkpoint with the highest CSI-M on a validation subset, using exponential moving average weights.

**General Performance (SEVIR & ARSO).** As shown in Table 3, Figure 2, Table 4 and Figure 3, FlowCast establishes a new state-of-the-art across both diverse datasets. On SEVIR, it achieves the highest overall CSI-M, FSS-M-P16, and HSS-M and the lowest CRPS, demonstrating superior probabilistic calibration. On ARSO, FlowCast outperforms all baselines in all metrics besides FAR-M. Notably, on SEVIR, while the probabilistic baseline CasCast achieves the highest CSI-P16-M, it suffers from a significantly higher FAR-M compared to FlowCast (0.383 vs. 0.325). This indicates

that FlowCast strikes a superior balance between detection sensitivity and precision, avoiding the tendency to over-predict precipitation coverage. Deterministic models achieve the lowest FAR-M scores by predicting blurry fields at longer lead times, missing the most extreme events (CSI-219).

**Performance on Extreme Events.** To evaluate the ability of the models to detect extreme events, we report performance at the highest intensity thresholds for both datasets in Table 5. FlowCast demonstrates a decisive advantage here. On SEVIR (Threshold 219), FlowCast achieves a CSI of 0.202, outperforming the best deterministic baseline (SimVP, 0.137) by over 47% and the leading probabilistic baseline (CasCast, 0.195). The trend holds for ARSO (Threshold 39 dBZ), where FlowCast achieves the highest CSI (0.183) and HSS (0.291). Crucially, FlowCast maintains this high detection skill while achieving a lower FAR than CasCast across all extreme thresholds, confirming its ability to generate sharp, intense features without resorting to excessive false alarms.

Table 3: Comparison of FlowCast with baseline models on the SEVIR dataset. All metrics are computed over a 12-step forecast, except "Forecast @ +65 min" which only uses the last frame.

| Model | CRPS ↓ | Forecast @ 12 steps | | | | | Forecast @ +65 min | |
| | | CSI-M ↑ | CSI-P16-M ↑ | FSS-P16-M ↑ | HSS-M ↑ | FAR-M ↓ | CSI-M ↑ | CSI-219 ↑ |
|---|---|---|---|---|---|---|---|---|
| U-Net | 0.0273 | 0.394 | 0.384 | 0.661 | 0.497 | 0.308 | 0.259 | 0.009 |
| Earthformer | 0.0252 | 0.411 | 0.407 | 0.686 | 0.518 | **0.285** | 0.280 | 0.016 |
| Earthfarseer | 0.0256 | 0.389 | 0.393 | 0.636 | 0.486 | 0.289 | 0.247 | 0.001 |
| SimVP | 0.0249 | 0.423 | 0.424 | 0.701 | 0.532 | 0.298 | 0.280 | 0.012 |
| PreDiff | 0.0189 | 0.413 | 0.423 | 0.699 | 0.523 | 0.313 | 0.281 | 0.018 |
| CasCast | 0.0201 | 0.442 | **0.520** | 0.763 | 0.562 | 0.383 | 0.311 | 0.054 |
| **FlowCast** | **0.0182** | **0.460** | 0.506 | **0.767** | **0.580** | 0.325 | **0.324** | **0.057** |

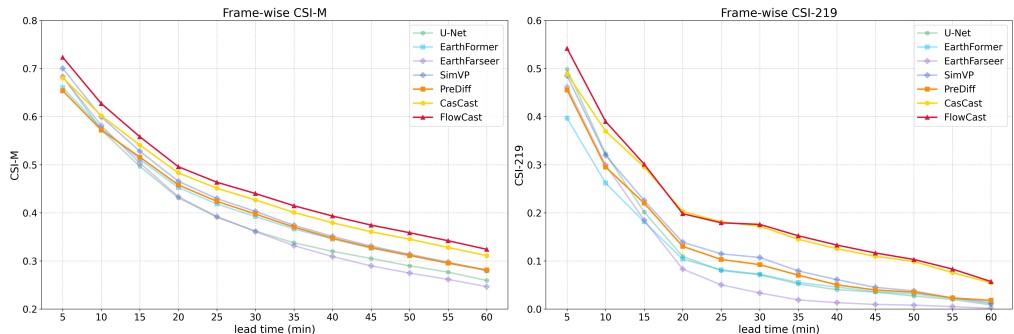

Figure 2: CSI-M and CSI at the 219 threshold per lead time on the SEVIR dataset. FlowCast shows consistent improvement over baselines for CSI-M and avoids the oversmoothing of deterministic models at longer lead times (CSI-219).

Table 4: Comparison of FlowCast with baseline models on the ARSO dataset. All metrics are computed over a 12-step forecast, except "Forecast @ +65 min" which only considers the last frame.

| Model | CRPS ↓ | Forecast @ 12 steps | | | | | Forecast @ +65 min | |
| | | CSI-M ↑ | CSI-P16-M ↑ | FSS-P16-M ↑ | HSS-M ↑ | FAR-M ↓ | CSI-M ↑ | CSI-39 ↑ |
|---|---|---|---|---|---|---|---|---|
| U-Net | 0.0264 | 0.399 | 0.432 | 0.659 | 0.505 | 0.371 | 0.260 | 0.011 |
| Earthformer | 0.0270 | 0.403 | 0.439 | 0.691 | 0.512 | 0.409 | 0.274 | 0.010 |
| Earthfarseer | 0.0280 | 0.368 | 0.406 | 0.588 | 0.463 | **0.358** | 0.233 | 0.004 |
| SimVP | 0.0267 | 0.415 | 0.462 | 0.699 | 0.526 | 0.401 | 0.288 | 0.029 |
| PreDiff | 0.0211 | 0.369 | 0.411 | 0.614 | 0.471 | 0.400 | 0.241 | 0.010 |
| CasCast | 0.0253 | 0.373 | 0.511 | 0.712 | 0.483 | 0.488 | 0.277 | 0.057 |
| **FlowCast** | **0.0209** | **0.420** | **0.514** | **0.738** | **0.535** | 0.422 | **0.315** | **0.073** |

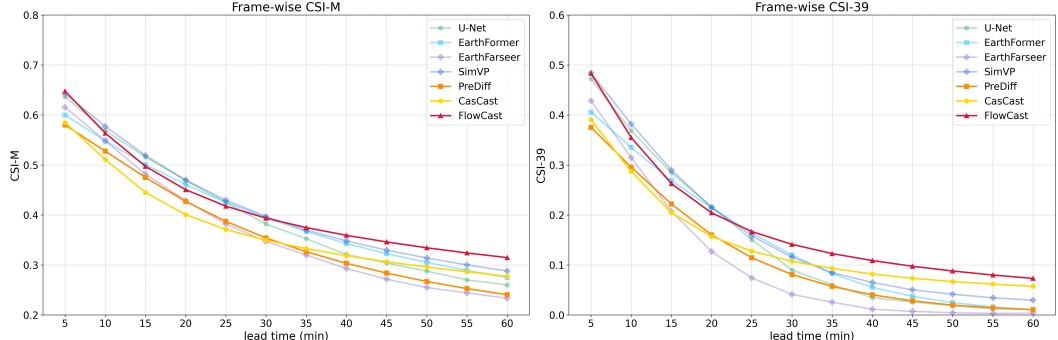

Figure 3: CSI-M and CSI at the 39 DBz threshold per lead time on the ARSO dataset. FlowCast shows significant improvements over probabilistic baselines for earlier lead times, and over deterministic baselines for later lead times.

Table 5: Comparison of FlowCast with baseline models on SEVIR and ARSO datasets for extreme events using categorical metrics with the highest thresholds per dataset. All metrics are computed over a 12-step forecast.

| Model | SEVIR | | | | | | ARSO | | | | | |
| | CSI | | HSS | | FAR | | CSI | | HSS | | FAR | |
| | 181 | 219 | 181 | 219 | 181 | 219 | 36 | 39 | 36 | 39 | 36 | 39 |
|---|---|---|---|---|---|---|---|---|---|---|---|---|
| U-Net | 0.205 | 0.122 | 0.314 | 0.193 | 0.366 | 0.508 | 0.209 | 0.145 | 0.318 | 0.226 | 0.474 | 0.505 |
| Earthformer | 0.229 | 0.109 | 0.348 | 0.180 | 0.354 | **0.343** | 0.216 | 0.145 | 0.335 | 0.231 | 0.531 | 0.553 |
| Earthfarseer | 0.194 | 0.097 | 0.291 | 0.152 | **0.341** | 0.412 | 0.159 | 0.104 | 0.245 | 0.164 | **0.445** | **0.449** |
| SimVP | 0.244 | 0.137 | 0.365 | 0.220 | 0.370 | 0.404 | 0.238 | 0.162 | 0.362 | 0.254 | 0.507 | 0.540 |
| PreDiff | 0.237 | 0.128 | 0.361 | 0.206 | 0.384 | 0.467 | 0.176 | 0.118 | 0.277 | 0.193 | 0.520 | 0.566 |
| CasCast | 0.286 | 0.195 | 0.427 | 0.309 | 0.501 | 0.567 | 0.202 | 0.142 | 0.320 | 0.235 | 0.647 | 0.694 |
| **FlowCast** | **0.301** | **0.202** | **0.443** | **0.317** | 0.425 | 0.482 | **0.254** | **0.183** | **0.388** | **0.291** | 0.547 | 0.589 |

Figure 4 qualitatively compares forecast sequences from FlowCast with the baselines on the SEVIR dataset. FlowCast produces sharp, perceptually realistic forecasts, avoiding the smoothness of deterministic models. Compared to the best-performing probabilistic baseline CasCast, we observe more realistic precipitation patterns, especially at longer lead times. More examples, including on ARSO, are provided in Appendix A.2.

## 4.3 ABLATION STUDIES

Due to computational constraints, all ablation studies were run on the first 10% of the SEVIR test set using a single NVIDIA A100 GPU.

### 4.3.1 CFM OBJECTIVE AGAINST DIFFUSION

To isolate the benefits of the CFM objective, we compare FlowCast against a strong baseline using the same backbone architecture but trained with a diffusion objective. We trained a DDPM (Ho et al., 2020) for 1000 timesteps. For efficient inference, we employed a DDIM sampler (Song et al., 2021) with a varying number of steps. This provides a strong and practical baseline to evaluate FlowCast against a highly optimized diffusion process on the same powerful architecture.

The results in Table 6 clearly demonstrate the superiority of the CFM objective. With a single step, FlowCast (CFM) drastically outperforms the DDIM sampler, even a 100-step DDIM baseline, across key metrics like CRPS and CSI-M, whilst being almost 100 times faster.

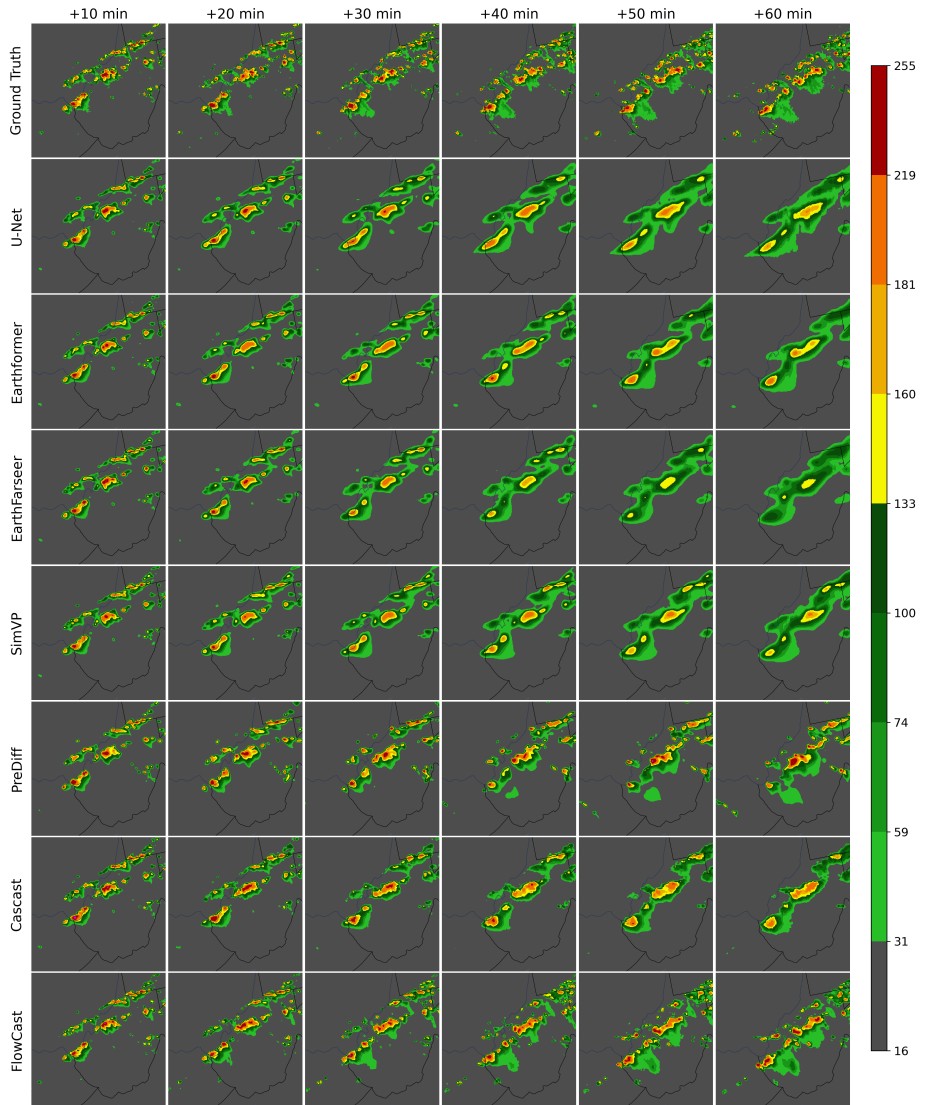

Figure 4: Qualitative comparison of FlowCast with other baselines on a SEVIR sequence. Columns show lead times from 10 to 60 minutes. Rows show the ground-truth, followed by the models.

Table 6: Ablation study: CFM vs. diffusion objective. Results highlight the superior performance and efficiency of the CFM framework. All metrics are computed over a 12-step forecast.

| Model | CRPS ↓ | CSI-M ↑ | CSI-P16-M ↑ | FSS-M-P16 ↑ | HSS-M ↑ | FAR-M ↓ | Time/Seq. (s) |
|---|---|---|---|---|---|---|---|
| CFM (1 steps) | 0.0207 | 0.454 | 0.504 | 0.763 | 0.571 | 0.337 | **2.6** |
| CFM (10 steps) | **0.0168** | **0.455** | **0.514** | **0.764** | **0.572** | 0.338 | 24 |
| DDIM (10 steps) | 0.0262 | 0.395 | 0.450 | 0.622 | 0.503 | 0.335 | 24 |
| DDIM (50 steps) | 0.0212 | 0.398 | 0.451 | 0.635 | 0.504 | 0.321 | 120 |
| DDIM (100 steps) | 0.0208 | 0.398 | 0.450 | 0.664 | 0.502 | **0.319** | 239 |

### 4.3.2 INFERENCE EFFICIENCY: PERFORMANCE VS. NUMBER OF FUNCTION EVALUATIONS

We assess inference efficiency by comparing FlowCast and the diffusion backbone across a range of function evaluations (NFE), where one NFE is an Euler (CFM) or DDIM step. Each NFE adds 2.4s per 8-member ensemble forecast. Figure 5 shows FlowCast is highly efficient, nearing optimal CRPS and CSI-M scores in just 3-10 steps. In contrast, the diffusion model requires 20-50 steps to peak and degrades sharply below 10 NFE. These results highlight the superior efficiency of the CFM framework, which learns a more direct mapping to the data manifold and enables high-fidelity forecasts with significantly fewer model evaluations. This efficiency is a crucial advantage for operational settings where forecasts must be both rapid and reliable.

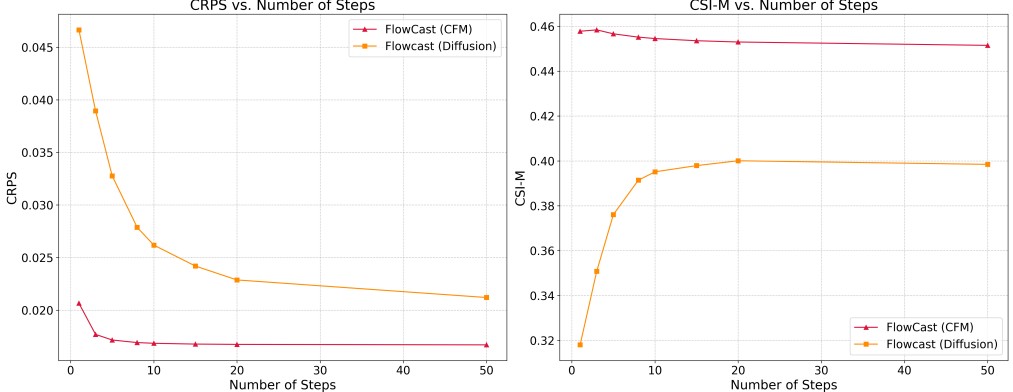

Figure 5: Performance vs. efficiency trade-off. Forecast quality (CRPS ↓, CSI-M ↑) as a function of NFE. FlowCast (CFM) achieves near-optimal performance with only 3 to 10 steps, while the DDIM-based model requires 20 steps to 50 steps, and degrades sharply at low NFE.

## 5 CONCLUSION

In this paper, we introduced FlowCast, the first fully probabilistic model applying Conditional Flow Matching (CFM) as a direct noise-to-data generative framework for precipitation nowcasting. Our experiments on the SEVIR and ARSO datasets show that FlowCast achieves state-of-the-art performance. Through direct ablation studies, we showed that the CFM objective is not only more accurate than a traditional diffusion objective on the same architecture but also vastly more efficient. FlowCast maintains high forecast quality with as few as a single sampling step, a regime where diffusion models fail. Our results firmly establish CFM as a powerful, efficient, and practical alternative for high-dimensional spatiotemporal forecasting.

**Limitations and Future Work:** While FlowCast shows significant promise, we identify two primary areas for future development. First, its reliance solely on radar data could be a limitation. Future work should explore multi-modal data fusion (e.g., satellite, NWP) to enhance robustness and accuracy. Second, our evaluation was limited to two datasets due to computational cost; a broader study across more meteorological regimes is needed to confirm generalizability.

ACKNOWLEDGMENTS

This work was supported by the Slovenian Research and Innovation Agency (ARIS) research core funding P2-0209 (Jana Faganeli Pucer). Special thanks go to Janko Merše and Dr. Matic Šavli from the Slovenian Environment Agency (ARSO) for their support and for providing the ARSO dataset.

REPRODUCIBILITY STATEMENT

To ensure the reproducibility of our work, we have provided comprehensive supporting materials. **Code:** The full source code for our FlowCast model, including scripts for training and evaluation, is available at `https://github.com/b-rbmp/FlowCast`. The repository contains detailed

instructions for setting up the required software environment and running the experiments. **Datasets:** Our work utilizes two datasets. The SEVIR dataset is a public benchmark, and details for access are provided by Veillette et al. (2020). The ARSO dataset was provided by the Slovenian Environment Agency (ARSO) for this research; we are actively collaborating with the agency to facilitate its public release in the near future. **Experimental Details:** Section 4.1 of the main paper provides a detailed description of our experimental setup. Further implementation details are available in Appendix A.1, including all model hyperparameters.

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

## A  APPENDIX

### A.1  IMPLEMENTATION DETAILS

This section details the implementation of FlowCast. The code to train and evaluate FlowCast is provided as supplementary material.

#### A.1.1  VAE

The VAE was configured with the hyperparameters shown in Table 7. Different warmup periods were empirically selected based on the convergence speed of the VAE on the respective datasets; ARSO converged faster due to less diversity in the data, attributed to factors such as its fixed geographical coverage and the significantly smaller stride used during sequence extraction.

Table 7: VAE hyperparameter summary

| Category | Parameter | Value / Setting |
|---|---|---|
| **Dataset** | | |
| | Source | SEVIR (vil) & ARSO (zm) |
| | Input Dimensionality (per frame) | $384 \times 384 \times 1$ (SEVIR), $301 \times 401 \times 1$ (ARSO) |
| | Input Preprocessing | Frame values scaled to $[0, 1]$ |
| **Training Objective** | | |
| | Loss Components | Reconstruction + KL Divergence + Adversarial |
| | KL Divergence Weight ($\lambda_{KL}$) | $1 \times 10^{-4}$ |
| | Discriminator Weight ($\lambda_{adv}$) | 0.5 |
| | Adversarial Loss Type | Hinge Loss |
| | Discriminator Architecture | PatchGAN (Isola et al., 2017) |
| | Discriminator Activation Warmup | 35 epochs (SEVIR), 15 epochs (ARSO) |
| **Optimization** | | |
| | Optimizer (Generator & Disc.) | AdamW |
| | Learning Rate (Initial) | $1 \times 10^{-4}$ |
| | Weight Decay | $1 \times 10^{-5}$ |
| | AdamW Betas | $(0.9, 0.999)$ |
| | LR Scheduler | Cosine Annealing with Linear Warmup |
| | LR Warmup Fraction | 20% of total training steps |
| | LR Min Warmup Ratio | 0.1 |
| | Min. LR Ratio | $10^{-3}$ |
| **Training Configuration** | | |
| | Batch Size | 12 (Global), 3 (Local) |
| | Max. Number of Epochs | 250 |
| | Gradient Clipping Norm | 1.0 |
| | Early Stopping Patience | 50 epochs |
| | Early Stopping Metric | Generator validation loss |
| | Training Nodes | $4 \times$ H100 GPUs |
| | FP16 Training | Disabled |
| **Model Configuration** | | |
| | Latent Channels | 4 |
| | GroupNorm Num | 32 |
| | Layers per Block | 2 |
| | Activation Function | SiLU |
| | Encoder-Decoder Depth | 4 |
| | Block Out Channels | [128, 256, 512, 512] |

**Data Preprocessing and Padding.** To accommodate the VAE's downsampling factor of $f = 8$, inputs must be spatially divisible by the downsampling rate. For the ARSO dataset, the native

resolution of $301 \times 401$ is not divisible by 8. We handle this by applying replication padding to the input frames to reach the nearest multiple of 16 prior to encoding. Specifically, the height is padded from 301 to 304, and the width from 401 to 416. This results in the latent dimensions of $38 \times 52$ reported in Table 2 (304/8 and 416/8, respectively). During inference, the generated fields are cropped back to the original $301 \times 401$ dimensions before evaluation metrics are computed. SEVIR dimensions ($384 \times 384$) are naturally divisible by 8, requiring no padding.

### A.1.2 FLOWCAST

**Architecture.** The FlowCast architecture, adapted from Earthformer-UNet (Gao et al., 2023) for latent-space Conditional Flow Matching, has the following configuration:

- **Core U-Net Architecture:**
  - **Hierarchical Stages:** A U-Net with 2 hierarchical stages (one level of downsampling/upsampling within the main U-Net body, in addition to initial/final processing).
  - **Stacked Cuboid Self-Attention Modules:** Each stage in both the contracting (encoder) and expansive (decoder) paths contains a depth of 4 Stacked Cuboid Self-Attention modules.
  - **Base Feature Dimensionality:** 192 units.

- **Spatial Processing:**
  - **Downsampling:** Achieved using Patch Merge (reducing spatial dimensions by a factor of 2 and doubling channel depth).
  - **Upsampling:** Uses nearest-neighbor interpolation followed by a convolution (halving channel depth).

- **Cuboid Self-Attention Details:**
  - **Pattern:** Follows an axial pattern, processing temporal, height, and width dimensions sequentially.
  - **Attention Heads:** 4 attention heads.
  - **Positional Embeddings:** Relative positional embeddings are used.
  - **Projection Layer:** A final projection layer is part of the attention block.
  - **Dropout Rates:** Dropout rates for attention, projection, and Feed-Forward Network (FFN) layers are set to 0.1.

- **Global Vectors:** The specialized global vector mechanism from the original Earthformer is disabled.

- **FFN and Normalization:**
  - **FFN Activation:** Feed-forward networks within the attention blocks use GELU activation.
  - **Normalization:** Layer normalization is applied throughout the relevant parts of the network.

- **Embeddings:**
  - **Spatiotemporal Positional Embeddings:** Added to the input features after an initial projection.
  - **CFM Time Embeddings ($t$):**
    * **Generation:** Generated with a channel multiplier of 4 relative to the base feature dimensionality (resulting in $192 \times 4 = 768$ embedding channels).
    * **Incorporation:** Injected into the network at each U-Net stage using residual blocks that fuse the time embedding with the feature maps (`TimeEmbedResBlock` modules).

- **Skip Connections:** Standard U-Net additive skip connections merge features from the contracting path to the expansive path.

- **Padding:** Zero-padding is used where necessary to maintain tensor dimensions during convolutions or cuboid operations.

**Training and Inference Hyperparameters.** The FlowCast training and inference hyperparameters are as follows:

Table 8: FlowCast hyperparameter summary

| Category | Parameter | Value / Setting |
|---|---|---|
| *Dataset* | | |
| | Source | Latent-space sequences from VAE |
| | Input Dimensionality | $13 \times 48 \times 48 \times 4$ (SEVIR), $13 \times 38 \times 52 \times 4$ (ARSO) |
| | Input Preprocessing | Standardized with training set statistics (mean, std) |
| *Training Configuration* | | |
| | Loss Function | MSE: $\mathcal{L} = \|\hat{v} - (Z_{\text{future}} - Z_P)\|^2$ |
| | Batch Size | 12 (Global), 3 (Local) |
| | Max. Number of Epochs | 200 |
| | Gradient Clipping Norm | 1.0 |
| | Early Stopping Patience | 50 epochs |
| | Early Stopping Metric | CSI-M evaluated on subset (40 batches) of validation set |
| | Training Nodes | $4 \times$ H100 GPUs |
| | FP16 Training | Enabled |
| | Exponential Moving Average Weights | Enabled |
| | Exponential Moving Average Weights Decay | 0.999 |
| *Optimization* | | |
| | Optimizer | AdamW |
| | Learning Rate (Initial) | $5 \times 10^{-4}$ |
| | Weight Decay | $1 \times 10^{-4}$ |
| | AdamW Betas | $(0.9, 0.999)$ |
| | LR Scheduler | Cosine Annealing with Linear Warmup |
| | LR Warmup Fraction | 1% of total training steps |
| | LR Min Warmup Ratio | 0.1 |
| | Min. LR Ratio | $10^{-2}$ |
| *CFM Parameters* | | |
| | $\sigma$ | 0.01 |
| | ODE Solver | Euler Method with 10 steps |

**Choice of ODE Solver.** We conducted an ablation study to compare various ODE solvers, including adaptive methods (Adaptive Heun, Dormand-Prince 5) and fixed-step methods (Euler, Midpoint, Runge-Kutta 4) on the first 10% of the SEVIR test set using a single NVIDIA A100 GPU. For adaptive solvers, a relative and absolute tolerance of $10^{-2}$ and $10^{-3}$ were used, respectively. Since no significant performance differences were observed, as shown in Table 9, we selected the Euler method with 10 steps for its computational efficiency and simplicity.

Table 9: Ablation study: ODE solvers. Results highlight minor differences in performance between the different solvers.

| Solver | CRPS ↓ | CSI-M ↑ | CSI-P16-M ↑ | FSS-M-P16 ↑ | HSS-M ↑ | FAR-M ↓ | Time/Seq. (s) |
|---|---|---|---|---|---|---|---|
| Euler (1 steps) | 0.0207 | 0.454 | 0.504 | 0.763 | 0.571 | 0.337 | 2.6 |
| Euler (10 steps) | 0.0168 | 0.455 | 0.514 | 0.764 | 0.572 | 0.338 | 24 |
| Dormand-Prince 5 | 0.0168 | 0.450 | 0.516 | 0.762 | 0.567 | 0.341 | 46 |
| Midpoint (10 steps) | 0.0167 | 0.451 | 0.516 | 0.762 | 0.568 | 0.341 | 44 |
| Runge-Kutta 4 (10 steps) | 0.0167 | 0.451 | 0.516 | 0.762 | 0.567 | 0.341 | 83 |
| Adaptive Heun | 0.0167 | 0.451 | 0.516 | 0.762 | 0.568 | 0.341 | 50 |

### A.1.3 Evaluation Metrics

**Continuous Ranked Probability Score (CRPS).** The CRPS is evaluated directly at the original data resolution, without applying any spatial pooling. For each ensemble of $N$ forecast members, CRPS is calculated at every pixel and then averaged across all spatial positions and forecast lead times to obtain a single summary metric. If the predictive distribution $F$ at a given pixel and time step is approximated by a Gaussian with mean $\mu$ and standard deviation $\sigma$ (estimated from the ensemble), and $x$ is the observed value, the CRPS can be computed as:

$$\text{CRPS}(F, x) = \sigma \left( \frac{x - \mu}{\sigma} \left( 2\Phi \left( \frac{x - \mu}{\sigma} \right) - 1 \right) + 2\phi \left( \frac{x - \mu}{\sigma} \right) - \frac{1}{\sqrt{\pi}} \right), \tag{1}$$

where $\Phi$ and $\phi$ denote the cumulative distribution function (CDF) and probability density function (PDF) of the standard normal distribution, respectively. Following standard practice, we normalize the CRPS for each dataset by its maximum values, which scales our reported metrics by 1/255 for SEVIR and 1/57 for ARSO.

**Threshold-based categorical metrics.** For each chosen intensity threshold $u$, we binarize the continuous ground truth field $x_{t,i,j}$ to obtain an observation mask $\mathbb{1}[x_{t,i,j} > u]$. The representative forecast $\hat{x}_{t,i,j}$ (see Section 4.1.2) is likewise thresholded to produce a binary forecast mask $\mathbb{1}[\hat{x}_{t,i,j} > u]$.

Using these binary masks and the dataset-specific thresholds, we construct a $2 \times 2$ contingency table for each evaluation:

|  | Observ. yes | Observ. no |
|---|---|---|
| Forecast yes | $H$ | $F$ |
| Forecast no | $M$ | $C$ |

$H$ : Hits (True Positives),
$M$ : Misses (False Negatives),
$F$ : False Alarms (False Positives),
$C$ : Correct Negatives.

The values $H, M, F, C$ are summed over all spatial locations, batches, and forecast lead times. The following metrics are then computed:

**False Alarm Ratio (FAR):** The proportion of forecasted events that did not actually occur.

$$\text{FAR} = \frac{F}{H + F}. \tag{2}$$

**Critical Success Index (CSI):** The fraction of observed and/or forecasted events that were correctly predicted, ignoring correct negatives. This metric is sensitive to both missed events and false alarms.

$$\text{CSI} = \frac{H}{H + M + F}. \tag{3}$$

**Heidke Skill Score (HSS):** The accuracy of the forecast relative to random chance.

$$\text{HSS} = \frac{2(HC - MF)}{(H + M)(M + C) + (H + F)(F + C)}. \tag{4}$$

**Pooled CSI (CSI-P16):** Forecasts and ground truth fields are first downsampled by applying a max-pooling operation over non-overlapping $16 \times 16$ pixel blocks. CSI is then recomputed on these pooled fields. A hit anywhere within a $16 \times 16$ block is registered as a success, thereby rewarding models that capture the local presence and intensity of precipitation, even if exact pixel-level alignment is imperfect.

**Fractions Skill Score (FSS):** We compute FSS to evaluate spatial alignment. Let $S_f(i, j)$ and $S_o(i, j)$ denote the fraction of pixels exceeding the threshold within a neighborhood of size $n \times n$ centered at $(i, j)$ for the forecast and observation, respectively. The FSS is calculated as:

$$\text{FSS} = 1 - \frac{\sum_{i,j}[S_f(i, j) - S_o(i, j)]^2}{\sum_{i,j} S_f(i, j)^2 + \sum_{i,j} S_o(i, j)^2} \tag{5}$$

## A.2 MORE QUALITATIVE EXAMPLES

This section presents additional qualitative examples comparing FlowCast against the baseline models

### A.2.1 SEVIR

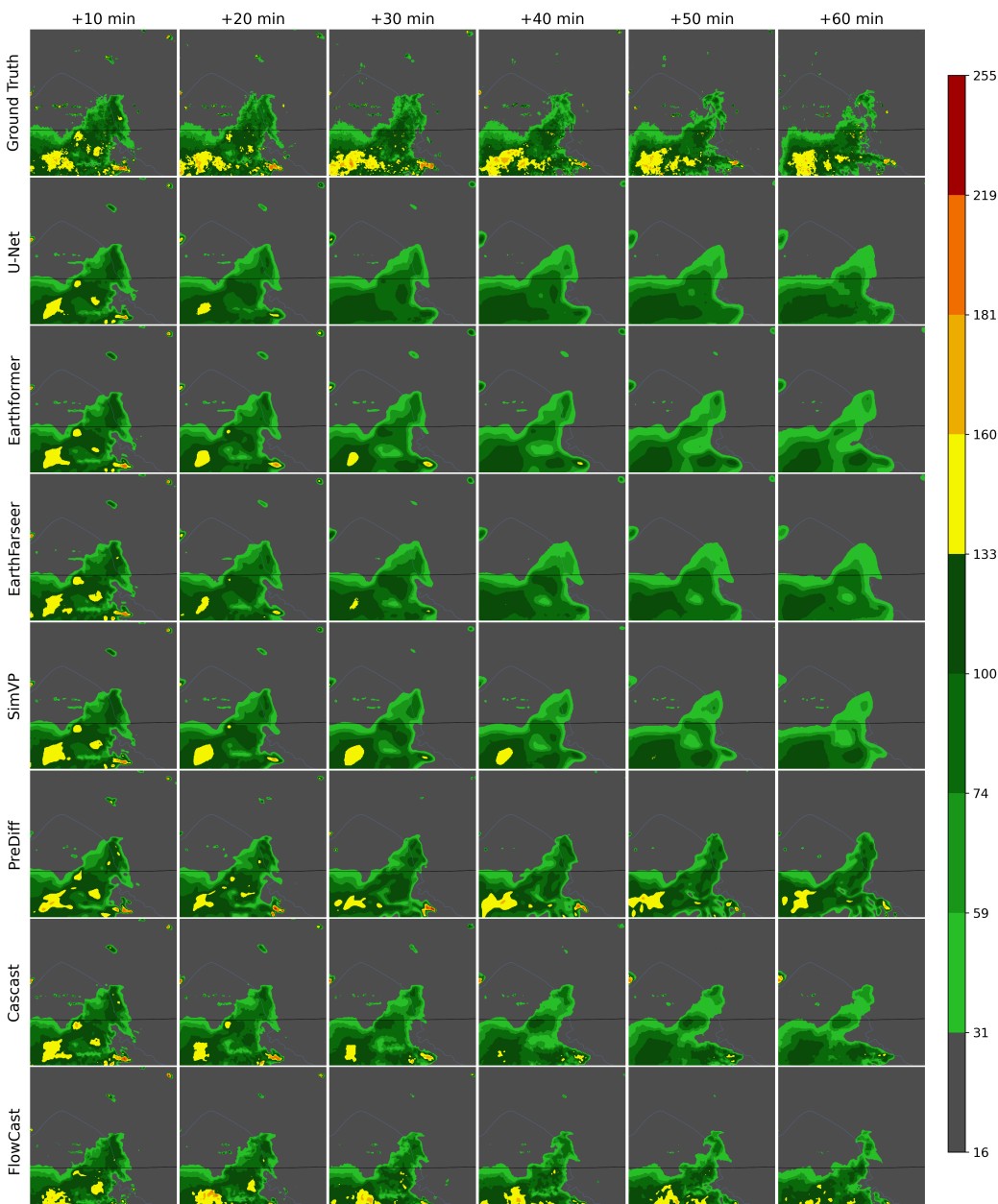

Figure 6: Qualitative comparison of FlowCast with other baselines on a SEVIR sequence.

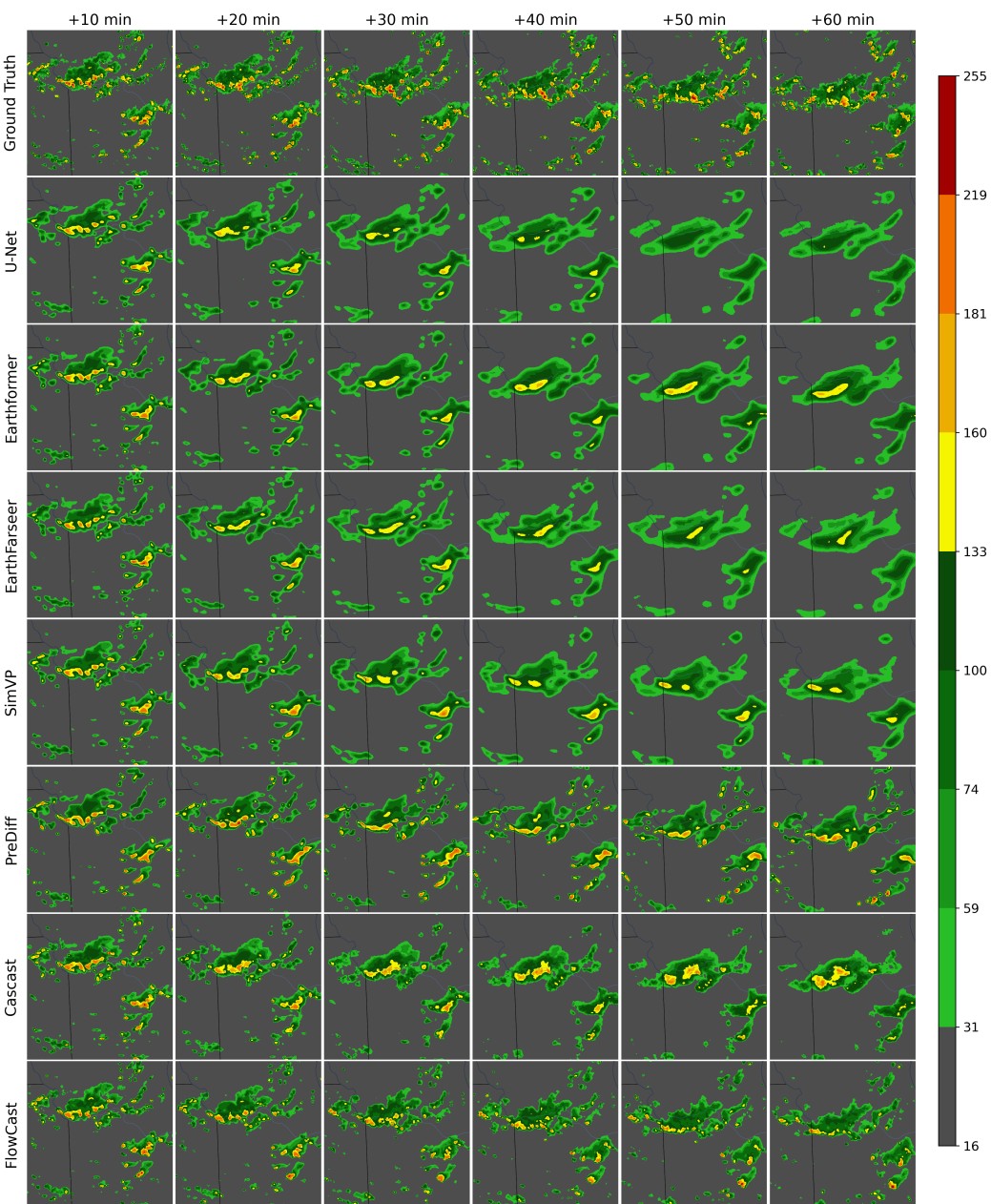

Figure 7: Qualitative comparison of FlowCast with other baselines on a SEVIR sequence.

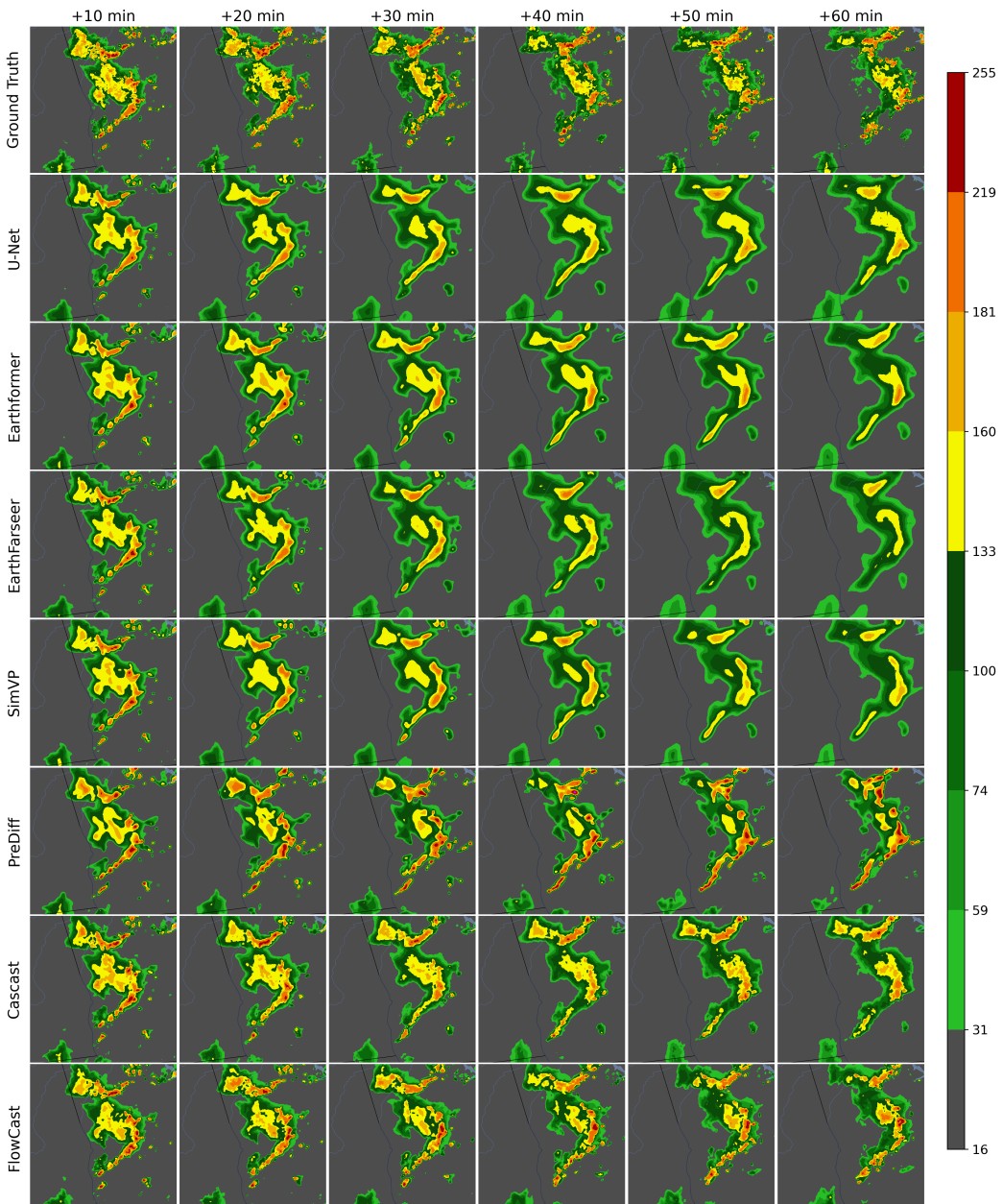

Figure 8: Qualitative comparison of FlowCast with other baselines on a SEVIR sequence.

## A.2.2   ARSO

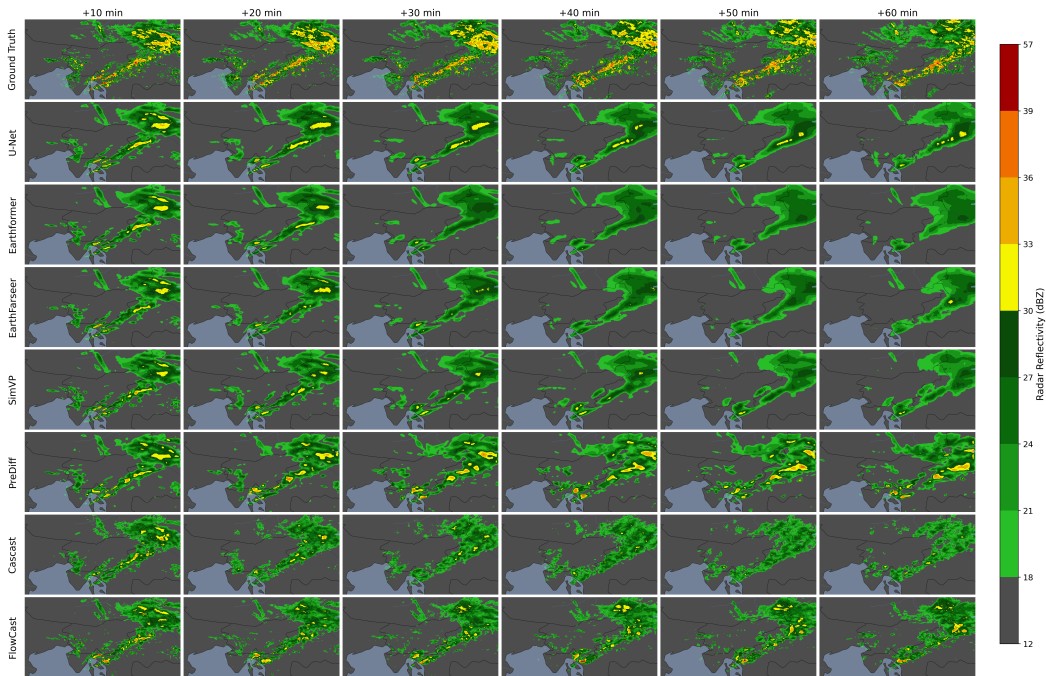

Figure 9: Qualitative comparison of FlowCast with other baselines on an ARSO sequence.

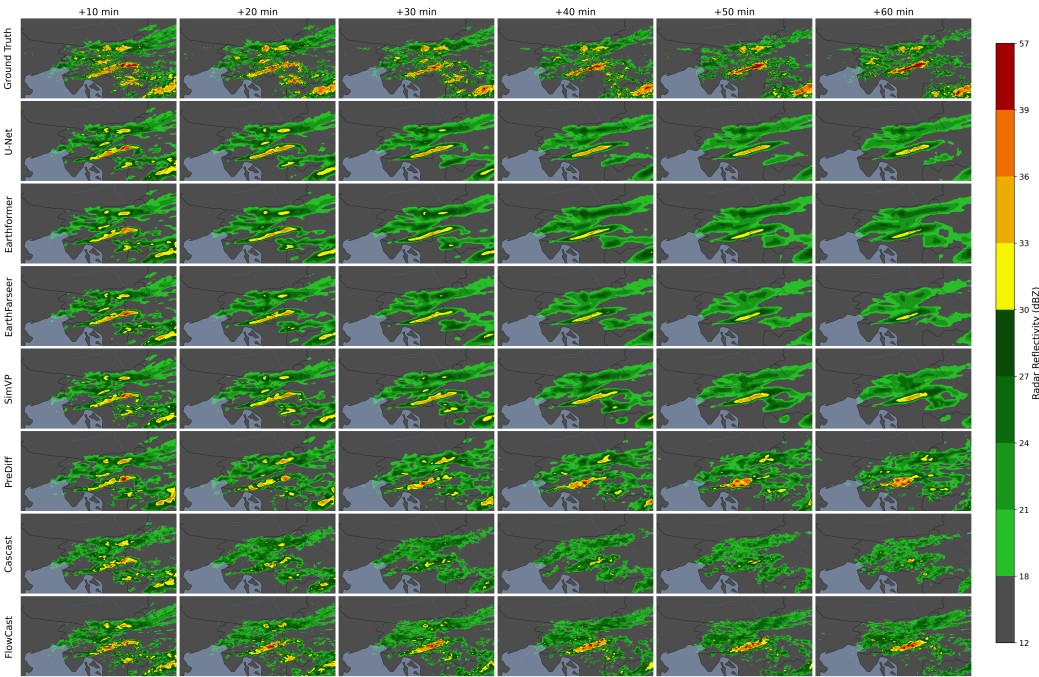

Figure 10: Qualitative comparison of FlowCast with other baselines on an ARSO sequence.

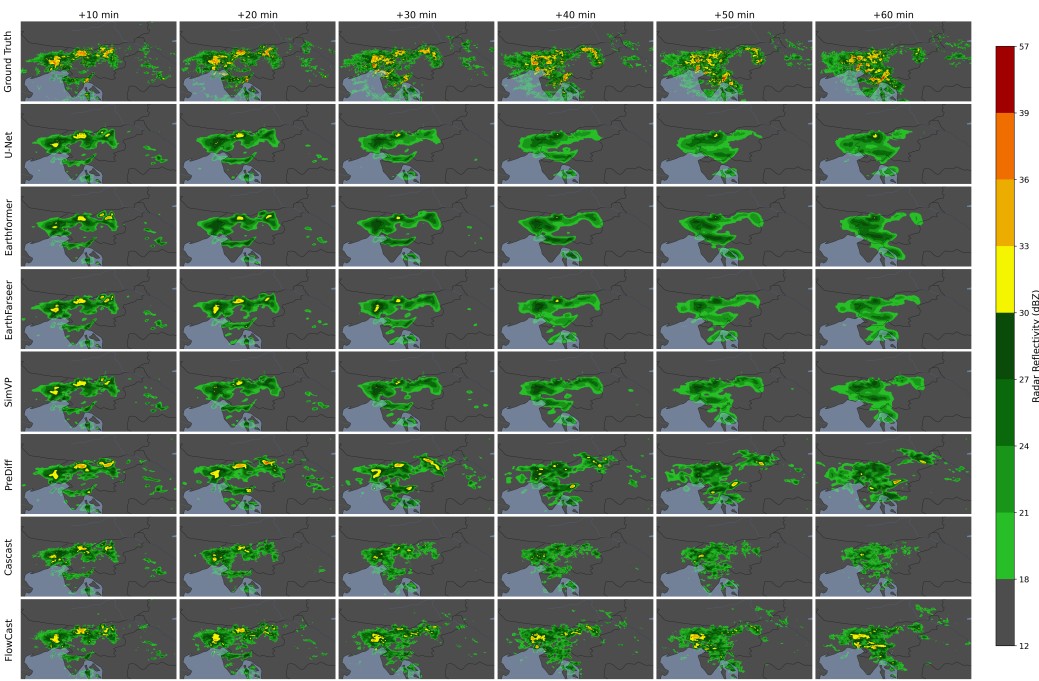

Figure 11: Qualitative comparison of FlowCast with other baselines on an ARSO sequence.

## A.3 LLM USAGE STATEMENT

During the preparation of this manuscript, we utilized a large language model (LLM) as a writing assistant. The LLM's primary role was to help refine sentence structure, improve clarity, and ensure grammatical correctness and consistency in tone. All scientific contributions, including the research ideation, methodological design, experimental analysis, and interpretation of results, were conducted solely by the authors, who take full responsibility for the content of this work.

