# OpenReview forum: "FlowCast: Advancing Precipitation Nowcasting with Conditional Flow Matching"
_ICLR.cc/2026/Conference — ICLR 2026 Poster_

### Official Review · Reviewer_zhbk · 2025-10-28

**Soundness:** 1
**Presentation:** 3
**Contribution:** 1
**Rating:** 2
**Confidence:** 4

**Summary:**

This paper introduces FlowCast, which applies Conditional Flow Matching (CFM) to radar-based precipitation nowcasting. The authors claim this is the first attempt to use CFM for spatiotemporal probabilistic forecasting. The model operates in a latent space via a VAE encoder-decoder and adopts an Earthformer-UNet backbone to learn a noise-to-data vector field. Experiments on the SEVIR and ARSO datasets show that FlowCast achieves state-of-the-art results in CRPS, CSI, and HSS, while requiring far fewer function evaluations than diffusion-based models. The paper highlights the superior efficiency–accuracy trade-off of CFM over diffusion objectives and positions FlowCast as a practical framework for real-time nowcasting.

**Strengths:**

S1. This paper demonstrates significantly reduced inference steps compared with diffusion methods, making it relevant for operational nowcasting systems that demand fast turnaround.

S2. FlowCast achieves superior or competitive scores on two diverse radar datasets, confirming generalization beyond a single domain.

S3. The paper provides explicit architectural and training settings, enhancing reproducibility.

S4. It addresses the real bottleneck in generative nowcasting (diffusion sampling inefficiency) with a concrete solution.

**Weaknesses:**

W1. The claim in the abstract and introduction that this work is the first to apply Conditional Flow Matching to precipitation nowcasting is inaccurate, since prior studies such as the ICML 2025 rectified flow model have already demonstrated flow-based precipitation refinement.

W2. The literature review in Section 2 on probabilistic nowcasting omits flow-based approaches when contrasting generative paradigms, resulting in an incomplete overview of probabilistic modeling beyond GANs and diffusion.

W3. The introduction of flow models in Section 2 does not provide a comparative analysis against other flow-based frameworks such as Continuous Normalizing Flows or rectified flows, leaving the advantages of CFM for spatiotemporal uncertainty unexplained.

W4. The architectural description in Section 3.2.2 shows that FlowCast merely adapts Earthformer-UNet to the CFM objective without introducing new architectural mechanisms or theoretical insights that would justify CFM’s suitability for atmospheric dynamics.

W5. The experimental comparison in Section 4.2 excludes recent transformer-based baselines such as Earthfarseer, which limits the credibility of the claimed state-of-the-art performance.

W6. The evaluation in Section 4.2 relies heavily on aggregated scores such as CSI-M and HSS-M, which mask variations across precipitation intensities and obscure the model’s behavior on extreme rainfall events.

W7. The methodological and inference descriptions in Sections 3.2.2 and 4.1.4 omit discussion of training instability and ODE-based sampling overhead, leaving open questions regarding robustness and computational scalability.

**Questions:**

1. The abstract mentions “predictive accuracy,” while the contributions section claims “probabilistic performance,” so which one defines the main contribution?
2. The description in Section 4.1.1 on Page 5 states that the ARSO dataset uses “63,716 training samples,” while Table 1 lists only 38,229 training samples. How can this internal inconsistency in dataset splits be explained?
3. Table 2 on Page 6 defines the ARSO latent dimensions as 13/12 × 38 × 52 × 4, which implies non-integer compression factors compared to the original 301 × 401 resolution. How is the rounding or padding handled to yield these dimensions?
4. The results table for ARSO on Page 8 labels the highest-threshold metric as “CSI-219,” even though Section 4.1.2 defines ARSO thresholds as [15, 21, 30, 33, 36, 39] dBZ. Thus, why does the same SEVIR-specific label appear in the ARSO table, and what threshold does it actually represent?
5. The conclusion on Page 9 asserts that FlowCast “maintains high forecast quality with as few as a single sampling step,” while all experiments use 10 steps for evaluation. How is this claim supported?
6. Appendix Table 6 uses different VAE warmup epochs for SEVIR and ARSO despite both using the same VAE architecture. What rationale justifies this discrepancy in the training configuration?
7.Appendix A.1.2 mentions an ODE solver ablation but gives no data. How is the claim of no difference supported?

---

> ### Author Response · Authors · 2025-11-22
> **Response to Reviewer zhbk - Part 1**
>
> We thank the reviewer for their rigorous and detailed assessment. We have addressed your concerns below, particularly regarding the novelty claim and the addition of the Earthfarseer baseline.
>
> ---
>
> ## Weaknesses
>
> ### W1 & W2: Novelty Claims and Flow-Based Literature
>
> We sincerely thank the reviewer for this crucial reference. We were unaware of this recent work and have revised our manuscript to cite Feng et al. (2025) [1] in our Related Work section and correct our novelty claim elsewhere. Specifically, we clarify that **FlowCast is the first end-to-end probabilistic CFM model for nowcasting**, distinguishing it from refinement-based approaches. The core difference lies in the generative process:
>
> - **Feng et al. (PercpCast)** operates as a **deterministic refinement model**. It uses a flow model to map a blurry forecast (from a deterministic ConvLSTM) to a sharp target.
> - **FlowCast** is a **full probabilistic generative model**. It learns a direct mapping from a simple noise prior $Z \sim \mathcal{N}(0,I)$ to the complex data distribution.
>
> Refinement methods are constrained by the modes captured by the initial deterministic model. If the initial "blurry mean" drops rare or extreme modes, the refinement step struggles to recover them. In contrast, our end-to-end approach captures the full multimodal distribution and tail statistics from scratch.
>
> While we investigated adding PercpCast as a quantitative baseline, it proved infeasible for this rebuttal due to three critical incompatibilities:
>
> 1. **Reproducibility:** The repository indicated in the ICML submission was empty at the time of rebuttal, and the paper does not specify several key architectural parameters (e.g., channel counts, latent dimension), making a faithful reproduction infeasible within the short rebuttal period.
> 2. **Resolution Mismatch:** Their method operates on downsampled 128×128 inputs for SEVIR, whereas FlowCast and standard benchmarks (e.g., Earthformer, SimVP, CasCast) operate on the original 384×384 resolution.
> 3. **Task Mismatch:** They evaluate on a non-standard 3-hour horizon (36 frames), differing from the standard 1-hour protocol used in the SEVIR Nowcasting Challenge and our work.
>
> [1] Feng, Wenzhi, et al. "Perceptually Constrained Precipitation Nowcasting Model." *Forty-second International Conference on Machine Learning*, 2025.
>
> ---
>
> ### W3: Comparison with Other Flow-Based Frameworks
>
> We appreciate this feedback. We have expanded **Section 2** to explicitly contrast CFM with traditional Continuous Normalizing Flows (CNFs) and Rectified Flows, clarifying the theoretical advantages of our approach:
>
> **vs. Traditional CNFs:**
> Standard CNFs (e.g., Neural ODEs, Chen et al., 2018) train via maximum likelihood, which requires solving an ODE (simulation) at every training step to compute the density change. For high-dimensional spatiotemporal data ($384 \times 384 \times 12$), this simulation loop is computationally intractable. CFM (Lipman et al., 2023) circumvents this by regressing a vector field against a pre-defined conditional probability path, enabling simulation-free training.
>
> **vs. Diffusion:**
> While diffusion models rely on stochastic denoising paths that are often curved and require many steps to integrate, CFM learns an ODE that approximates a straight-line trajectory between noise and data (Tong et al., 2024). We argue this provides a better inductive bias for precipitation nowcasting, enforcing a direct mapping that preserves temporal coherence, and enables the high-efficiency sampling (1 to 10 steps) demonstrated in our results.
>
> **vs. Rectified Flows:**
> We acknowledge that I-CFM (Tong et al., 2024) and Rectified Flows (Liu et al., 2023) share the simulation-free training paradigm. However, our usage of I-CFM with a Gaussian probability path ($\sigma>0$) offers a distinct advantage over the standard 1-Rectified Flow. Using $\sigma>0$ produces a Gaussian-smoothed probability path, which stabilizes vector-field regression by ensuring non-zero support around the interpolation trajectory, preventing overfitting to singular paths.

---

> ### Author Response · Authors · 2025-11-22
> **Response to Reviewer zhbk - Part 2**
>
> ### W4: Justification for CFM's Suitability for Atmospheric Dynamics
>
> We argue that the suitability of CFM for atmospheric dynamics is demonstrated by specific theoretical and computational advantages:
>
> **Theoretical Advantage:**
> We have revised **Section 1 (Introduction)** to explicitly articulate that CFM offers a superior inductive bias via geometric simplification. Radar reflectivity distributions are highly multi-modal yet exhibit strong local temporal consistency. Standard diffusion models often traverse "winding" stochastic paths to map noise to this complex manifold, requiring many steps to resolve fine-grained structures. In contrast, CFM imposes a straight-line ODE prior that enforces the most direct mapping between noise and data distributions. This explicitly preserves temporal coherence and stability, allowing FlowCast to approximate complex dynamics efficiently without the trajectory curvature inherent to diffusion.
>
> **Computational Advantage:**
> We further highlight that the efficiency of the CFM objective enables the Earthformer-UNet backbone to scale to resolutions previously intractable for diffusion. When this architecture was originally introduced for probabilistic nowcasting in PreDiff (Gao et al., 2023), the high computational cost of diffusion training necessitated downsampling the SEVIR data to $128 \times 128$, requiring **32 days to train on four NVIDIA A10G GPUs** [1]. In contrast, FlowCast's simulation-free CFM training enabled us to train on the full-resolution ($384 \times 384$) SEVIR data in just **7 days (on 4×H100)**. Thus, the contribution is demonstrating that CFM unlocks the full capacity of spatiotemporal transformers that were previously bottlenecked by the diffusion objective.
>
> [1] Official Implementation for PreDiff: https://github.com/gaozhihan/PreDiff
>
> ---
>
> ### W5: Missing Earthfarseer Baseline
>
> As noted in our general response, we have conducted additional experiments training Earthfarseer (Wu et al., 2024) on the SEVIR dataset, following the official GitHub implementation. In our setup, Earthfarseer did not outperform the SimVP or Earthformer baselines (**see table in General Comment**). We will add Earthfarseer as a baseline for all tables, plots and comparisons to the manuscript as soon as training on ARSO finishes, which will be done before the end of the rebuttal period.
>
> This was also a result found in the recent rectified flow nowcasting paper published at ICML 2025, where it only leads in MSE but lags behind SimVP and Earthformer among most other metrics [1].
>
> [1] Feng, Wenzhi, et al. "Perceptually Constrained Precipitation Nowcasting Model." *Forty-second International Conference on Machine Learning*, 2025.
>
> ---
>
> ### W6: Evaluation Reliance on Aggregated Scores
>
> Thank you for this observation. We agree that aggregated scores can mask performance on the tail of the distribution. As highlighted in our **General Comment**, we have addressed this by adding a **dedicated Extreme Event Analysis** in the revised **Section 4.2**, where we explicitly partition the evaluation to isolate high-intensity thresholds
>
> As shown in our new results, FlowCast achieves the **highest CSI and HSS scores** on the most severe thresholds across both SEVIR (CSI-181, CSI-219) and ARSO (CSI-36, CSI-39). Furthermore, FlowCast establishes a superior performance trade-off compared to the leading probabilistic baseline, CasCast. We not only achieve higher detection scores (CSI/HSS) but do so with a **significantly lower False Alarm Ratio**, demonstrating that FlowCast generates sharper, more accurate extreme forecasts with fewer hallucinations.
>
> ---
>
> ### W7: Training Instability and Sampling Overhead
>
> **Sampling Overhead:**
> We measured this: the per-step cost of CFM and DDIM is identical (2.4s per 8-member ensemble - **Table 6**). Since we are using Euler's method, each additional step simply adds one forward pass to the model. In the newly added "ODE Solvers" ablation study (**Table 9, Appendix**) and now in **Table 6**, we show that even running a single CFM Euler step results in strong results, outperforming DDIM with 100 steps and confirming its scalability.
>
> **Training Instability:**
> We added a note addressing this point to **Sec 4.1.3**. Unlike the complex, weighted losses of diffusion, the I-CFM objective is a simple, stable regression (MSE-like) loss, which we found converged robustly without extensive tuning.
>
> ---

---

> ### Author Response · Authors · 2025-11-22
> **Response to Reviewer zhbk - Part 3**
>
> ## Questions
>
> ### Q1: Predictive Accuracy vs. Probabilistic Performance
>
> We thank the reviewer for this opportunity to clarify our contribution. We consider both aspects to be central to FlowCast's value proposition:
>
> - **Probabilistic Performance:** As a generative model, FlowCast captures the full distribution of possible futures better than diffusion baselines (achieving the lowest CRPS).
> - **Predictive Accuracy:** We demonstrate that this superior distributional modeling also yields state-of-the-art deterministic accuracy (CSI, HSS) when evaluating the ensemble mean, surpassing deterministic models like Earthformer.
>
> To reflect this dual advantage, we have updated both the **Abstract** and the **Contributions section** to explicitly state that FlowCast establishes a new state-of-the-art in probabilistic performance, while simultaneously exceeding deterministic baselines in predictive accuracy.
>
> ---
>
> ### Q2: ARSO Training Sample Inconsistency
>
> This was a typo in the text description. The value in **Table 1 (38,229)** is the correct number of training samples, and 63,716 is the total number of samples in the entire dataset. We have corrected the text.
>
> ---
>
> ### Q3: ARSO Latent Dimensions and Padding
>
> We thank the reviewer for noting this detail. The non-integer compression arises because the ARSO dataset dimensions (301×401) are not divisible by the VAE's downsampling factor of 8.
>
> We apply replication padding to the inputs prior to encoding. Specifically, we pad the input spatial dimensions to the nearest multiple of 16:
> - Height: 301 → 304
> - Width: 401 → 416
>
> The padded input (304×416) is then passed through the VAE encoder, resulting in the latent dimensions reported in **Table 2**. During inference, the VAE decoder outputs the padded resolution (304×416), and we crop the output back to the original 301×401 resolution before computing metrics. We have added this clarification to **Appendix A.1.1** in the revised manuscript.
>
> ---
>
> ### Q4: CSI-219 Label in ARSO Table
>
> Thank you for this observation. CSI-219 is a SEVIR-specific label. In the ARSO table (**Table 4**), this column represents **CSI-39 dBZ** (the highest threshold) which roughly corresponds to the exceedance probability of 219 in SEVIR. We have corrected the table header.
>
> ---
>
> ### Q5: Single Sampling Step Claim
>
> While this result was previously implicit in the efficiency plot (**Figure 5**, where the FlowCast curve begins at a high performance level), we have now explicitly added this data to **Table 6** to support the claim.
>
> This confirms that FlowCast significantly outperforms the diffusion (DDIM) baseline on the same architecture, even when the latter is allowed 100 steps. This demonstrates that the CFM objective learns a sufficiently straight probability path to allow for accurate one-step generation, whereas the diffusion baseline collapses at low NFE. While we use 10-step FlowCast (where CRPS stabilizes) for our main results, the model produces very coherent structures at 1 step, unlike diffusion.
>
> ---
>
> ### Q6: Different VAE Warmup Epochs
>
> We have clarified this in **Appendix A.1.1** of the revised paper. The difference was empirically selected based on the convergence characteristics of the VAE on each dataset.
>
> The ARSO dataset represents a fixed geographical domain and was processed with a high overlap (stride 1), resulting in lower visual diversity and higher data redundancy. Consequently, the VAE converged to a stable reconstruction quality significantly faster than on SEVIR, which contains diverse weather events across the continental US with lower redundancy (stride 12). We therefore reduced the warmup period for ARSO (15 epochs) compared to SEVIR (35 epochs) to match these convergence rates.
>
> ---
>
> ### Q7: ODE Solver Ablation Data
>
> We apologize for the omission. We have added **Table 9 (Appendix A.1.2)** which details the ablation. It shows negligible performance differences between different ODE methods, including the fixed-step Euler (10 steps), Midpoint (10 steps), and Runge-Kutta 4 (10 steps) and the adaptive Dormand-Prince 5 and Adaptive Heun, justifying our choice of the simplest approach (Euler).
>
> ---
>
> We hope these clarifications and the new baseline results adequately address your concerns and we sincerely thank you for all the points raised in your review.

---

> > ### Comment · Reviewer_zhbk · 2025-11-27
> >
> > I appreciate the authors' detailed response and the substantial revisions made to the manuscript. In particular, the authors have effectively redefined the model's novelty and contributions, while providing comprehensive theoretical explanations and additional experiments to substantiate the appropriateness and advantages of the chosen architecture. However, I note a minor formatting inconsistency in the revised manuscript's Section 2 (RELATED WORK): the subsections "Deterministic Nowcasting" and "Probabilistic Nowcasting" should be standardized, either both formatted as subheadings or neither. Overall, these enhancements have significantly improved the quality of the paper, despite some remaining minor flaws. As a result, I am updating my rating to 4 (marginally below the acceptance threshold. But would not mind if paper is accepted).

---

> > > ### Author Response · Authors · 2025-12-02
> > > **Response to Reviewer zhbk**
> > >
> > > We sincerely thank the reviewer for their continued engagement, positive feedback, and for raising the score. We are glad to hear that our revisions regarding the model's novelty and theoretical justification addressed your core concerns.
> > >
> > > Regarding the formatting inconsistency in Section 2, we have corrected this in the latest revision. We have standardized the structure by creating a formal **"2.1. Deterministic Nowcasting"** subsection to match the formatting of **"2.2. Probabilistic Nowcasting."**
> > >
> > > Thank you again for helping us improve the quality of our manuscript.

---

### Official Review · Reviewer_EpeH · 2025-10-30

**Soundness:** 3
**Presentation:** 3
**Contribution:** 2
**Rating:** 4
**Confidence:** 4

**Summary:**

The proposed FlowCast model is the first to apply Conditional Flow Matching to precipitation nowcasting, which utilizes VAE to compress radar data into latent space, combines Cuboid Attention and CFM to learn direct noise-data mapping for efficient sampling. On SEVIR and ARSO datasets, its various metrics outperform baselines like PreDiff and CasCast, with 10-step sampling more efficient than diffusion models.

**Strengths:**

1. This work is the first to apply Conditional Flow Matching to precipitation nowcasting, learning a direct noise-to-data mapping. It achieves high-fidelity forecasting with only 10 sampling steps.
2. This work adopts VAE latent space compression and Cuboid Attention architecture, balancing the efficiency of high-dimensional radar data processing and the ability of spatiotemporal dynamic modeling.

**Weaknesses:**

1. Conditional Flow Matching achieves the mapping from noise to data by learning a vector field. However, the paper does not design a prior structure for the vector field in combination with the physical laws of precipitation, relying entirely on data-driven learning. This may lead the model to generate results lacking physical condition constraints.
2. This work applies CFM to precipitation nowcasting for the first time, but the idea of CFM has already been validated in some tasks within the field of computer vision. The paper transfers it to the precipitation nowcasting scenario and fails to propose improvement strategies for CFM targeting certain characteristics of precipitation data (such as strong spatiotemporal correlation, dynamic evolution, etc.). This results in deficiencies in innovation and generative capability.

**Questions:**

1. This work only validates performance on two radar datasets (SEVIR, ARSO). Are there experimental results tested on more datasets? Meanwhile, is the performance tested under shorter or longer lead times?
2. The probabilistic baselines compared in this work only include diffusion models. Is there a comparison with GAN-based probabilistic models?

---

> ### Author Response · Authors · 2025-11-22
> **Response to Reviewer EpeH**
>
> We thank the reviewer for their thoughtful feedback regarding physical constraints and experimental scope.
>
> ---
> ### W1 & W2: Lack of Physical Priors and CFM Improvements for Precipitation Data
>
> FlowCast follows the prevailing paradigm in modern deep learning nowcasting (e.g., Earthformer, CasCast), which relies on the model's capacity to **implicitly learn complex physical dynamics from large-scale data**.
>
> While explicit priors ensure adherence to specific laws, recent literature suggests they often incur a performance trade-off. For instance, PreDiff (Gao et al., 2023) demonstrates that explicit "Knowledge Alignment" degrades generative quality across datasets:
>
> - On N-body MNIST, imposing physical laws worsened the FVD score from 0.987 → 4.063
> - On the SEVIR precipitation dataset, the constrained model achieved a worse FVD (34.18) compared to the unconstrained baseline (33.05)
> - The authors note that such constraints can lead to "unrealistic forecast hallucinations" when the model struggles to satisfy imposed priors with scarce data
>
> To empirically verify that our data-driven approach learns plausible dynamics without these drawbacks, we conducted an additional Fractions Skill Score (FSS) analysis. As mentioned in our General Comment, the results on SEVIR confirm that **FlowCast captures spatial structure and coherence better than baselines**, successfully approximating physically plausible dynamics implicitly. Full results (including evaluation on ARSO) will be added to the manuscript before the end of the rebuttal period.
>
> ---
>
> ### Q1: Limited Dataset Validation and Forecast Horizons
>
> **1. Dataset Scope:**
> We agree that validating on additional datasets is always beneficial. However, we limited our scope to two diverse datasets (SEVIR and ARSO) primarily due to the **high computational cost** associated with training and sampling high-resolution generative models. As noted in our implementation details, a single FlowCast model requires approximately **7 days on 4 NVIDIA H100 GPUs** to train. This computational burden is also why our ablation studies were conducted on a 10% subset of the data.
>
> We believe that evaluating on both:
> - A large-scale public benchmark (SEVIR) with diverse US weather events
> - A distinct local Alpine dataset (ARSO) with complex orographic effects
>
> provides a robust validation of FlowCast's generalization across different meteorological regimes.
>
> **2. Lead Times:**
> Regarding the forecast horizon, we strictly adhered to the **standard SEVIR Nowcasting Challenge protocol** (65 minutes context, 60 minutes prediction, in 5-minute steps). We chose this specific setup to ensure a direct and fair comparison with the extensive body of existing literature (e.g., Earthformer, PreDiff, CasCast).
>
> ---
>
> ### Q2: Missing GAN-Based Probabilistic Baselines
>
> We thank the reviewer for this suggestion. Our focus was on comparing FlowCast to the **current SOTA probabilistic paradigm, which is diffusion-based** (PreDiff, CasCast). The previous SOTA, GAN-based models like DGMR, are no longer the primary performance benchmark.
>
> Recent literature such as the PreDiff paper (Gao et al., 2023), which we do compare against, has already established a significant performance gap over prior GAN-based models (e.g., PreDiff Table 2 shows superior CRPS and FVD vs. DGMR). Therefore, comparing against PreDiff effectively positions FlowCast relative to both the GAN-based and diffusion-based approaches.
>
> ---
>
> We appreciate your suggestions for strengthening our work and hope these clarifications address your concerns.

---

### Official Review · Reviewer_itD7 · 2025-10-31

**Soundness:** 2
**Presentation:** 2
**Contribution:** 2
**Rating:** 4
**Confidence:** 4

**Summary:**

The paper describes the new solution to the problem of radar-based precipitation nowcasting, which uses conditional flow matching (CFM). The authors highlight accuracy and efficiency based on the empirical evidence.

**Strengths:**

Significance: improving the performance and democratising access to precipitation forecasting is an important problem as addressed in this paper.

Clarity: the paper is in general clearly written

Quality: the methodology is sound, the description seems correct and reproducible. The ablation studies show the justification for experimental decisions.

**Weaknesses:**

Quality: The main argument is that swapping the training procedure from the diffusion model to conditional flow matching improves the performance, both in terms of accuracy and in terms of computational efficiency. The problem I see with this argument, however, is that there is previous evidence, in other domains, that the CFM may improve upon the efficiency and performance [1].  I would expect in this case for the authors to link it to the background and then say something that goes beyond the existing literature.  . In other words, the specific link to the justification of this method is missing.

Originality: as the narrative goes now, and stemming from the previous point, there is a concern that the work is an application of CFM for the precipitation forecasting. That should, in my mind, include a justification that is unique to the precipitation nowcasting (theoretical or otherwise).  Therefore, I would invite the authors to expand upon the contribution and say how this work goes beyond the existing literature in this aspect.

In summary, my main concern is that the claims now appear to be a combination of the two: CFM and precipitation forecasting.  I would invite the authors therefore to expand upon why this reasoning could be wrong.


[1] Lipman et al (2023) Flow Matching  for Generative Modeling, ICLR 2023

**Questions:**

1. I would appreciate if the authors add confidence intervals to the key results (Tables 3-5 in particular).

2. The algorithms for training and sampling at Page 4 are difficult to follow, therefore I would suggest that the authors present them as algorithm blocks.

3. The authors say in the abstract: "the uncertainty of atmospheric dynamics and the efficient modeling of high-dimensional data" I am curious whether this paper addresses this question in any way beyond the accuracy metrics?

---

> ### Author Response · Authors · 2025-11-22
> **Response to Reviewer itD7**
>
> We sincerely thank the reviewer for their insightful comments on the domain-specific justification for CFM and the clarity of our presentation.
>
> ---
>
> ### W1: Domain-Specific Justification for CFM
>
> We agree that simply applying a method known to be efficient in other domains is insufficient without a domain-specific justification. In our revision, we have **expanded the introduction** to clarify that **CFM offers a superior inductive bias for precipitation nowcasting** due to its ability to simplify the transport of probability mass in this specific high-dimensional domain.
>
> Our justification rests on the relationship between the multi-modal nature of radar data and the geometry of the generative path:
>
> - **Radar reflectivity distributions** are highly multi-modal yet exhibit strong local temporal consistency (the state at time $t$ is highly correlated with $t+1$).
>
> - **Standard diffusion models** rely on stochastic denoising or curved probability flow ODEs to map Gaussian noise to this complex data manifold. These trajectories are often highly non-linear and "winding," necessitating many sampling steps to resolve the fine-grained structure of precipitation without blurring the modes.
>
> - **CFM imposes a straight-line ODE prior** on the generative process. This enforces the simplest possible mapping between the noise distribution and the data distribution. In the context of spatiotemporal forecasting, where temporal coherence is essential, this linear interpolation of probability mass provides a much stronger and more stable prior than the diffusion process.
>
> I-CFM encourages a vector field aligned with the straight probability path, producing trajectories that are empirically close to linear.
>
> Therefore, the contribution is showing that for the specific topology of precipitation data, this **trajectory linearization is a major enabler of operational efficiency**. As shown in our ablation (**Figure 5**), this geometric simplification allows FlowCast to maintain high fidelity and temporal consistency with as few as **1–10 function evaluations**, whereas the curved paths of diffusion models degrade and collapse at this low regime.
>
> ---
>
>
> ### Q1: Confidence Intervals
>
> We agree that confidence intervals are ideal. However, given the high computational cost of training generative video models (**7 days on 4×H100s** for FlowCast), training multiple instances for statistical intervals was not feasible within the rebuttal period.
>
> We rely on the consistency of our results across two distinct datasets (SEVIR and ARSO) and multiple metrics to demonstrate robustness. The fact that FlowCast consistently outperforms baselines across different geographical regions, meteorological regimes, and evaluation metrics provides strong evidence of its reliability.
>
> ---
>
> ### Q2: Algorithm Presentation
>
> Thank you for the suggestion. We have **reformatted the training and sampling algorithms** and the old training/sampling diagram into **standard algorithmic blocks** (**Algorithms 1 & 2**) in the revision for better readability.
>
> ---
>
> ### Q3: Uncertainty and High-Dimensional Modeling
>
> We address these claims through specific metrics and validations beyond standard accuracy:
>
> **1. Uncertainty:**
> We quantify "uncertainty" using the **Continuous Ranked Probability Score (CRPS)** (**Tables 3 & 4**), which evaluates the full predicted distribution rather than a single outcome. FlowCast achieves the **lowest CRPS (0.0182 on SEVIR)**, proving it effectively models the stochastic, multimodal nature of precipitation compared to deterministic or diffusion baselines.
>
> **2. Efficiency:**
> We address high-dimensional modeling by compressing inputs into a compact latent space (**384×384 → 48×48**) and validate computational cost in **Figure 5**. Our ablation demonstrates that FlowCast achieves optimal performance with only **3–10 function evaluations (NFE)**, reducing the computational burden by **an order of magnitude** compared to diffusion models.
>
> ---
>
> We appreciate your constructive feedback and hope our revisions adequately address your concerns.

---

### Official Review · Reviewer_25PR · 2025-11-01

**Soundness:** 2
**Presentation:** 3
**Contribution:** 2
**Rating:** 4
**Confidence:** 4

**Summary:**

This paper introduces FlowCast, which for the first time applies the efficient Conditional Flow Matching (CFM) framework to radar-based precipitation nowcasting. FlowCast integrates a variational autoencoder (VAE) for latent representation with a network backbone based on the Earthformer-UNet architecture. The authors aim to address the slow sampling issue inherent in existing diffusion models and use ablation studies to demonstrate the efficiency advantage of the CFM objective over the diffusion objective when applied to the same backbone.

**Strengths:**

1. **Innovation and Practicality of the Methodology:**
    - The paper innovatively introduces the highly efficient CFM generative framework to the high-dimensional spatiotemporal prediction problem of precipitation nowcasting. This represents a valuable exploration of cutting-edge methods in the field.

    - A direct comparison and ablation study with diffusion models clearly demonstrates the significant advantage of CFM in terms of efficiency, showing that it can maintain or closely approach optimal performance with very few function evaluations (NFE). This is of high practical value for time-sensitive nowcasting applications.

2. **Clarity and Readability:** The paper is well-structured, the methodology is detailed, and the explanation of the CFM training and sampling process is particularly concise and clear, making the working principle of FlowCast easy to understand.

3. **Recognition of Effort and Contribution:** Given that the authors performed comparisons across two generative paradigms (CFM vs. Diffusion) and conducted experiments on two diverse geographical and climatic datasets, they clearly put effort into advancing model performance and exploring efficiency.

**Weaknesses:**

1. **Incomplete Experimental Results and Lack of Convincing Evidence:**

    - **Inadequate Metric Presentation:** The experimental evaluation lacks quantitative verification of structured forecasts. For instance, the absence of analysis using spatial verification methods (e.g., FSS at different scales) prevents a thorough validation of the model's ability to predict the structure and location of precipitation systems. Furthermore, the authors did not provide complete performance curves for **all** baseline models across all future lead times on both datasets.

2. **Ambiguity in Innovation Positioning and Insufficient Exploration Breadth:**

    - **If positioned as Framework Innovation:** Simply applying the existing CFM training/sampling framework to a UNet backbone based on Earthformer Blocks results in a relatively low degree of methodological novelty for an ICLR-level conference.

    - **If positioned as Application Exploration:** The paper aims to explore the general applicability of CFM as an efficient generative paradigm for precipitation nowcasting, but the breadth of exploration is insufficient. The lack of comparative experiments applying CFM to at least one different type of advanced nowcasting model makes it difficult to strongly demonstrate CFM's effectiveness as a general objective function.

**Questions:**

**Regarding False Alarm Ratio (FAR) and Forecast Quality:** Although FlowCast achieves the highest HSS-M score, your results indicate a **higher FAR-M** (compared to Earthformer and PreDiff). This suggests that FlowCast's high-scoring forecasts may be accompanied by a high rate of 'false alarms'. Could you provide and discuss more detailed categorical skill scores, such as the Threat Score (TS) or Equitable Threat Score (ETS) across different precipitation thresholds? This would help provide a fairer assessment of the model's actual skill in predicting high-intensity precipitation events.

---

> ### Author Response · Authors · 2025-11-22
> **Response to Reviewer 25PR**
>
> We sincerely thank you for your constructive review and positive feedback. We are addressing your concerns with new experiments (FSS analysis) and clarifications, which we detail below.
>
> ---
>
> ### W1: Incomplete Experimental Results and Inadequate Metric Presentation
>
> **1. On Spatial Verification Metrics (e.g., FSS):**
>
> We agree that quantifying the model's ability to predict the structure and location of precipitation is critical. While our initial use of CSI-P16-M aimed to capture localized events, we acknowledge that the Fractions Skill Score (FSS) provides a more robust and standardized spatial validation.
>
> Per your suggestion, we have conducted FSS analysis across multiple spatial scales. The results on SEVIR (see our **General Comment**) indicate that **FlowCast achieves the highest FSS across the 1×1, 4×4, and 16×16 spatial scales**, confirming its superior structural accuracy. Full results, including the ongoing evaluation on ARSO, will be added to the manuscript by the end of the rebuttal period.
>
> **2. On Complete Performance Curves:**
>
> We apologize for any confusion regarding the performance curves:
>
> - **SEVIR:** The lead-time performance curves for SEVIR are currently available in **Figure 2** of the manuscript.
> - **ARSO:** While we included aggregated metrics in **Table 4**, we acknowledge the curves were missing. We have generated the corresponding lead-time curves for the ARSO dataset (CSI-M and CSI-39) and added them to the revision (**Figure 3**).
>
> ---
>
> ### W2: Ambiguity in Innovation Positioning
>
> We position FlowCast primarily as a **paradigm shift in application** rather than a purely architectural novelty.
> Current fully probabilistic SOTA models rely on diffusion, which suffers from a severe efficiency bottleneck. Our work is the first to demonstrate that an **end-to-end CFM objective substantially alleviates this bottleneck in precipitation nowcasting**. Our direct ablation study in **Table 6** isolates the objective function on the exact same backbone (Earthformer-UNet). The results show that CFM is:
>
> 1. **One order of magnitude more efficient** than diffusion
> 2. **Achieves superior metrics** across the board
>
> This empirical validation of noise-to-data CFM as a superior engine for high-dimensional weather forecasting is our core contribution.
>
> ---
>
> ### Q1: Threat Score (TS) and Equitable Threat Score (ETS)
>
> We clarify that the **Threat Score (TS) is mathematically identical to the Critical Success Index (CSI)** under their standard definitions in meteorological forecasting, which we have already reported extensively (**Tables 3 & 4**, **Figures 2 & 3**). Across these metrics, FlowCast consistently outperforms all baselines.
>
> Regarding the Equitable Threat Score (ETS), under the standard definitions used in our paper (as in Hogan et al., 2010 [1]), **ETS is a monotonic transformation of HSS**. Using those definitions, a model with higher HSS necessarily yields higher ETS.
>
> We prioritized reporting HSS because it is a **"truly equitable" metric**, meaning it awards a score of exactly zero to random forecasts regardless of sample size. In contrast, Hogan et al., 2010 [1] identifies ETS as only **"asymptotically equitable"**, noting that it is biased (yielding positive scores for random guesses) in finite sample sizes.
>
> Therefore, our **state-of-the-art HSS results implicitly demonstrate state-of-the-art ETS performance**, while utilizing a metric that is more robust to sample-size variations.
>
> [1] Hogan, R. J., C. A. T. Ferro, I. T. Jolliffe, and D. B. Stephenson, 2010: Equitability Revisited: Why the "Equitable Threat Score" Is Not Equitable. *Wea. Forecasting*, 25, 710–726, https://doi.org/10.1175/2009WAF2222350.1.
>
> ---
>
> ### Q2: FAR vs. Forecast Quality
>
> You correctly noted our FAR is higher than deterministic models like Earthformer. Our new **Extreme Event analysis (Table 5)** clarifies this trade-off:
>
> 1. **Deterministic models (Earthformer)** achieve low FAR (e.g., 0.343 on SEVIR-219) by predicting blurry fields that rarely cross high-intensity thresholds. However, this results in a **poor ability to detect severe weather** (CSI-219: 0.109).
>
> 2. FlowCast preserves variance and sharpness. While this incurs a higher FAR-219 (0.482), it **drastically improves the detection of actual events**, achieving a CSI-219 of 0.202 (an 85% improvement over Earthformer).
>
> 3. Crucially, compared to CasCast (the leading probabilistic competitor), FlowCast achieves:
>    - Higher CSI: 0.202 vs 0.195
>    - Significantly lower FAR: 0.482 vs 0.567
>
>    This demonstrates that we offer the best trade-off between precision and detection in the probabilistic class.
>
> ---
>
> We hope these clarifications and new metrics address your concerns, and we thank you again for helping us strengthen our paper.

---

### Author Response · Authors · 2025-11-22
**General Response to All Reviewers**

We sincerely thank all reviewers for their constructive feedback, which has substantially strengthened our manuscript. We have made significant revisions addressing the concerns raised, including:

---

## New Experimental Results

### 1. Extreme Event Forecasting Analysis
Addressing concerns that aggregated metrics mask model behavior on rare events, we isolated high-intensity precipitation thresholds (36 and 39 dBZ for ARSO, 181 and 219 for SEVIR) to compare performance specifically on severe weather. **This is now present in the revised manuscript.**

**Extreme Event Performance (Table 5 in revised manuscript):**

| Model | SEVIR CSI-181 | SEVIR CSI-219 | SEVIR HSS-181 | SEVIR HSS-219 | SEVIR FAR-181 | SEVIR FAR-219 | ARSO CSI-36 | ARSO CSI-39 | ARSO HSS-36 | ARSO HSS-39 | ARSO FAR-36 | ARSO FAR-39 |
|-------|--------------|--------------|--------------|--------------|--------------|--------------|-------------|-------------|-------------|-------------|-------------|-------------|
| U-Net | 0.205 | 0.122 | 0.314 | 0.193 | 0.366 | 0.508 | 0.209 | 0.145 | 0.318 | 0.226 | 0.474 | 0.505 |
| Earthformer | 0.229 | 0.109 | 0.348 | 0.180 | 0.354 | 0.343 | 0.216 | 0.145 | 0.335 | 0.231 | 0.531 | 0.553 |
| SimVP | 0.244 | 0.137 | 0.365 | 0.220 | 0.370 | 0.404 | 0.238 | 0.162 | 0.362 | 0.254 | 0.507 | 0.540 |
| PreDiff | 0.237 | 0.128 | 0.361 | 0.206 | 0.384 | 0.467 | 0.176 | 0.118 | 0.277 | 0.193 | 0.520 | 0.566 |
| CasCast | 0.286 | 0.195 | 0.427 | 0.309 | 0.501 | 0.567 | 0.202 | 0.142 | 0.320 | 0.235 | 0.647 | 0.694 |
| **FlowCast (ours)** | **0.301** | **0.202** | **0.443** | **0.317** | **0.425** | **0.482** | **0.254** | **0.183** | **0.388** | **0.291** | **0.547** | **0.589** |

### 2. ARSO Lead-Time Curves
We have generated the missing CSI-M and CSI-39 lead-time performance curves for ARSO. **This is now present as Figure 3 in the revised manuscript.**

### 3. Fractions Skill Score (FSS) Analysis
We conducted FSS evaluations at multiple spatial scales (1×1, 4×4, 16×16). Results on the SEVIR dataset are provided below. We are currently running inference for all baselines on the ARSO dataset, and both will be included in the final manuscript revision before the end of the rebuttal period.

**FSS Results (SEVIR):**

| Model | FSS-1-M | FSS-4-M | FSS-16-M |
|-------|---------|---------|----------|
| U-Net | 0.509 | 0.569 | 0.661 |
| Earthformer | 0.530 | 0.592 | 0.686 |
| SimVP | 0.543 | 0.607 | 0.701 |
| Earthfarseer | 0.498 | 0.554 | 0.636 |
| PreDiff | 0.535 | 0.600 | 0.699 |
| CasCast | 0.577 | 0.651 | 0.763 |
| **FlowCast (Ours)** | **0.590** | **0.664** | **0.767** |

*Note: FSS-4-M denotes 4×4 spatial scale, averaged over all SEVIR thresholds.*

### 4. Earthfarseer Baseline
We have trained and evaluated Earthfarseer on SEVIR. Results (shown below) indicate it underperforms SimVP and Earthformer on most metrics. We are currently completing training on ARSO, and all comparisons (tables, figures) will include Earthfarseer in the final revision.

**Earthfarseer Results (SEVIR):**

| Model | CRPS ↓ | CSI-M ↑ | CSI-P16-M ↑ | HSS-M ↑ | FAR-M ↓ |
|-------|--------|---------|-------------|---------|---------|
| Earthformer | 0.0252 | 0.411 | 0.407 | 0.518 | 0.285 |
| SimVP | 0.0249 | 0.423 | 0.424 | 0.532 | 0.298 |
| Earthfarseer | 0.0256 | 0.389 | 0.393 | 0.486 | 0.289 |

### 5. ODE Solver Ablation
We have added an ablation study showing no impact on performance with different choices of adaptive and fixed-step solvers (Appendix Table 9).

---

## Manuscript Revisions

**Section-Level Changes:**
- **Section 1 (Introduction):** Expanded with domain-specific justification for CFM's geometric advantages
- **Section 2.1 (Probabilistic Nowcasting):** Enhanced with explicit comparison to CNFs and Rectified Flows
- **Section 4.1.3 (Training Details):** Added discussion of training stability
- **Section 4.2 (State-of-the-Art Comparison):** Completely restructured to explicitly separate general performance from extreme event forecasting
- **Appendix A.1.1:** Added explanations for different VAE warmup periods and ARSO padding details
- **Appendix A.1.2:** Added ODE solver ablation (Table 9)

**Content Updates:**
- Corrected novelty claims to acknowledge Feng et al. (ICML 2025) and clarify FlowCast as the first end-to-end probabilistic CFM model (updated in Abstract, Introduction, and Conclusion)
- Fixed textual inconsistencies identified by Reviewer zhbk (ARSO sample count, table references, etc.)
- Reformatted training and sampling algorithms into standard algorithm blocks (Algorithms 1 & 2)
- Added CFM (1 step) row to ablation Table 6 to explicitly show efficiency gains over diffusion

---

We address specific concerns from each reviewer below.

---

### Author Response · Authors · 2025-12-02
**Final Update to All Reviewers and Area Chair**

We sincerely thank all reviewers for their time and constructive feedback, which has played a crucial role in strengthening our manuscript.

We are saddened that the recent system incident and subsequent reversion prevented us from receiving final responses from most reviewers. We remain confident that the comprehensive revisions detailed below, and in our previous General Response, have addressed the core concerns regarding baselines and extreme event performance.

In particular, we would like to express our gratitude to **Reviewer zhbk**. Prior to the system reversion, they acknowledged our updates and explicitly stated their decision to **raise their score to 4**, noting that the enhancements had "significantly improved the quality of the paper."

### Completion of Promised Experiments
In our previous update, we noted that certain experiments were ongoing. We confirm that **the final manuscript revision** now contains the full, completed set of results:

* **Full FSS Analysis (SEVIR & ARSO):** We have added the Fractions Skill Score evaluations for **both** datasets across the $16\times16$ spatial scale. These results confirm FlowCast's structural superiority across both datasets.
* **Earthfarseer Baseline:** We have finished training Earthfarseer on ARSO. It has now been added to **all** comparative tables and performance plots in the revised manuscript. Consistent with the SEVIR results shown previously, FlowCast outperforms this baseline on both aggregated and extreme-event metrics.
* **Formatting & Textual Polish:** We have implemented the specific formatting requests from Reviewer zhbk (standardizing Section 2 headings) and streamlined text throughout the paper for clarity.

Please refer to our **previous General Response (below)** for a detailed breakdown of the major revisions, including the **Extreme Event Analysis** and **Efficiency/Ablation studies**.

---

### Meta-Review · Area_Chair_WhFd · 2025-12-30

**Summary:**

This paper proposes an application of conditional flow matching (CFM) for the task of precipitation nowcasting. Reviewers agree that the approach is sound, well-executed, and shows promising empirical results.

The main weaknesses raised by the reviewers include:
- [W1] Insufficient metrics and analysis of the proposed approach (25PR, itD7, EPeH, zhbk)
- [W2] Limited novelty or incremental results (25PR, itD7, EPeH, zhbk)

**Reviewer Concerns:**

- [W1] The rebuttal successfully addresses most of the comments raised from the reviewers regarding baselines, metrics, and analyses.
- [W2] This is a deeper criticism that was largely not resolved in the rebuttal period.

**Reviewer Scores:**

- 25PR, itD7, EpeH may have raised their score slightly given the discussion and additional results. However, given the criticisms regarding novelty, they seem unlikely to raise their scores significantly.
- zhbk explicitly states that they would raise their score to 4.


Overall, the reviewers find the experiments be well-executed and the performance of the method to be competitive. However, there is agreement that the proposed approach largely represents an application of an existing method to new datasets. Further domain-specific insights would significantly strengthen the contributions of the paper. That being said, I think the paper is sufficiently strong in its current form to recommend acceptance.

---

### Decision · Program_Chairs · 2026-01-26

Accept (Poster)